# A single WNT enhancer drives specification and regeneration of the *Drosophila* wing

Elena Gracia-Latorre [1,4], Lidia Pérez [1,3,4], Mariana Muzzopappa [1] & Marco Milán [1,2] ✉

Wings have provided an evolutionary advantage to insects and have allowed them to diversify. Here, we have identified in *Drosophila* a highly robust regulatory mechanism that ensures the specification and growth of the wing not only during normal development but also under stress conditions. We present evidence that a single wing-specific enhancer in the *wingless* gene is used in two consecutive developmental stages to first drive wing specification and then contribute to mediating the remarkable regenerative capacity of the developing wing upon injury. We identify two evolutionary conserved cis-regulatory modules within this enhancer that are utilized in a redundant manner to mediate these two activities through the use of distinct molecular mechanisms. Whereas Hedgehog and EGFR signalling regulate Wingless expression in early primordia, thus inducing wing specification from body wall precursors, JNK activation in injured tissues induce Wingless expression to promote compensatory proliferation. These results point to evolutionarily linked conservation of wing specification and regeneration to ensure robust development of the wing, perhaps the most relevant evolutionary novelty in insects.

The evolutionarily conserved Wnt/β-catenin pathway is classically known for its fundamental role in governing key developmental decisions during embryonic development[1,2]. However, in the last two decades, it has been shown that this pathway is also used by vertebrates and invertebrates to drive cell proliferation upon tissue injury[3–6], and that its chronic activation is relevant in the development of cancer[7]. Identifying the molecular mechanisms by which the expression of Wnt ligands is regulated in space and time will contribute to the understanding of their roles in regulating developmental decisions, driving regeneration or causing cancer. Gene expression is governed mainly by enhancers, which are regions of non-coding DNA that recruit transcriptional factors and fully activate the transcription of a gene by interaction with its promoter[8]. The spatio-temporal expression of genes is determined by the activation of specific pathways in response to particular inputs plus the

accessibility of the enhancer. *Drosophila* Wingless (Wg) is the founding member of the Wnt family. It was not until 1973 that wingless flies were found in a stock by serendipity, and this mutation, named *wingless¹* (*wg¹*), was mapped to the second chromosome[9]. Despite the highly pleiotropic effects of Wg in the development of most fly tissues (reviewed in ref. [10]), the phenotype of *wg¹* flies was remarkably restricted to the absence of wings.

In this work, we present evidence that the *wg¹* phenotype is due to the loss of a highly conserved 1.8-kb-long enhancer, which is functionally restricted to the developing wing primordium. In young larvae, signalling molecules present in the wing primordium act on this enhancer to drive the expression of *wg* to the most distal part, where it induces wing fate specification. The combination of CRISPR/Cas9-mediated deletions and reporter assays allowed us to functionally identify and narrow down the wing-specific enhancer to two highly

[1]Institute for Research in Biomedicine (IRB Barcelona), The Barcelona Institute of Science and Technology, Baldiri Reixac, 10, 08028 Barcelona, Spain. [2]Institució Catalana de Recerca i Estudis Avançats (ICREA), Pg. Lluís Companys 23, 08010 Barcelona, Spain. [3]Present address: Hubrecht Institute, University Medical Centre Utrecht, Utrecht, The Netherlands. [4]These authors contributed equally: Elena Gracia-Latorre, Lidia Pérez. ✉e-mail: marco.milan@irbbarcelona.org

evolutionarily conserved *cis*-regulatory modules (CRMs) that are used in a redundant manner by Hedgehog and Vein, a ligand of the Epidermal Growth Factor receptor (EGFR), to mediate wing fate specification. In older larvae and once wing specification is underway, this enhancer remains silent but accessible to transcription factor binding in growing wing primordia[11,12]. Upon injury, this enhancer is activated by JNK signalling to drive regeneration[13]. We present evidence that the two CRMs present in the wing-specific enhancer are activated in a redundant manner not only in injured tissues to mediate Wg expression and wing regeneration but also in malignant tumours to drive unlimited proliferation. The use of a single wing-specific enhancer of the *wg* gene to drive wing specification and regeneration in two consecutive developmental stages unravels highly linked evolutionary conservation of these two activities. The functional redundancy of the two existing CRMs in mediating these two processes unveils a highly robust mechanism to guarantee proper wing development.

## Results

### A genomic region required for wing specification

In 1973, Dr. Sharma observed that the phenotype of *wg¹* mutant flies was markedly restricted to the absence of wings and was associated with a duplication of the notum, the dorsal side of the body wall that arises from the same larval primordium, the wing disc[9]. Fourteen years later, *wg¹* was shown to be linked to a small deficiency downstream of the *wg* gene[14] (Fig. 1a and named hereafter as Δ*wg¹* for simplicity). Unfortunately, though, it was not certain that this deletion was responsible for the mutant phenotype. The penetrance of the *wg¹* phenotype was not complete and only 40% of the heminota (half of a notum) lacked the appendage[15]. Similar results were recently obtained with the same allele, but penetrance was shown to be even lower (below 20%[13]). Given the observed reduction in the penetrance of the phenotype over time, we backcrossed the Δ*wg¹*-carrying chromosome twice into the *w¹¹¹⁸* genetic background to remove potential genetic suppressors. To characterise the potential contribution of the region deleted in the *wg¹* allele to the adult wingless phenotype, we crossed *wg¹* mutant flies with flies carrying either an independently generated overlapping deletion (Δ*BRV118*[13], see Fig. 1a) or larger chromosomal deficiencies (Supplementary Table 1). The penetrance of the wingless phenotype was over 90% in most transheterozygous conditions over large deficiencies and, most interestingly, it was up to 80% in Δ*BRV118*/Δ*wg¹* heterozygous flies (Fig. 1b, c). We observed the presence of a duplicated notum (nt) in all cases (Fig. 1c).

Wg is expressed in a highly dynamic manner in the developing larval wing disc (Figs. 1d and 2a). Early in development (in second instar, L2), Wg is expressed in a ventral anterior wedge (Figs. 1d and 2a, ref. 16), where it signals locally to specify the progenitors of the adult wing[17]. In third instar (L3), wing discs contain the progenitors of the adult wing and notum, and Wg expression at this developmental stage promotes wing growth and notum patterning, respectively (Figs. 1d and 2a, refs. 18–21). Interestingly, Δ*BRV118*/Δ*wg¹* individuals lacked Wg expression in second instar wing discs and the resulting mature wing discs showed loss of wing progenitors and duplication of notal precursors (Fig. 1d). The *wg¹*-associated deletion is placed between two *wnt* genes, namely *wg* and *wnt6*, and we observed that *wnt6* was expressed in the same pattern as *wg*, not only in late third instar wing discs[22,23], where it potentiates Wg-driven tissue growth[18], but also in second instar wing discs (Fig. 1e). However, several observations indicate that Wg, but not Wnt6, is the main ligand driving wing fate specification. Transheterozygous conditions of Δ*BRV118* or Δ*wg¹* with *wgᶜˣ⁴* (a null allele of *wg* deleting the first coding exon, ref. 24, shown in Fig. 1a) or with *wgᶜˣ³* (a 17kb–long insertion allele of *wg* located between the *wg¹*-enhancer and the *wg* gene, ref. 24, depicted in Fig. 1a) gave rise to adult flies lacking wings (Fig. 1b). We found that the penetrance was lower than Δ*BRV118*/Δ*wg¹* or Δ*BRV118* homozygous flies, most probably because of potential interactions in trans

(transvection) between the *wg¹*-enhancer and the *wg* promoter of homologous chromosomes. By contrast, adult flies homozygous for *wnt6ᴷᴼ* (a null allele of *wnt6* lacking the first coding exon, Fig. 1a) and transheterozygous flies of *wnt6ᴷᴼ* over large chromosomal deficiencies developed wings (Fig. 1b). Remarkably, ectopic expression of Wg, but not Wnt6, was able to induce secondary wings in the notum (Supplementary Fig. 1a). All these results indicate that the genomic region deleted in the *wg¹* mutation is required in early wing discs to drive Wg expression and wing fate specification, and they point to the presence of a wing-specifying enhancer in this region.

### The *wg¹* deletion contains an early enhancer

We next used a transgenic reporter gene assay to address whether the genomic region deleted in the *wg¹* mutation was sufficient to drive *lacZ* reporter expression in a pattern that reproduced the early pattern of Wg expression in the ventral anterior wedge of the wing disc. A *lacZ* reporter containing the genomic region deleted in the *wg¹* mutation (*wg¹-lacZ*) reproduced the early expression pattern of Wg in the ventral anterior wedge of the wing disc (Fig. 2e). However, this expression pattern persisted in older wing discs, although at lower levels, thereby suggesting that the regulatory elements responsible for turning it off during normal development are not present in this region. Similar results were obtained with a *lacZ* reporter containing the overlapping *BRV118* region (Fig. 2f). By contrast, a *lacZ*-enhancer trap placed in the promoter of the *wg* gene (Fig. 2b) recapitulated the dynamic expression pattern of Wg in the wing disc during larval development (Fig. 2d). These results indicate that the genomic region deleted in the *wg¹* mutation contains an enhancer that drives the characteristic expression of Wg in second instar wing discs.

In early wing discs, Hedgehog (Hh) coming from posterior (P) cells induces Wg expression in anterior (A) cells[17] (Fig. 2c). Consistent with this, we identified several bioinformatically predicted binding sites of the transcription factor Ci, the most downstream component of the Hh signalling pathway (Gli in vertebrates), in the *wg¹* genomic region (green bars in Fig. 2b, see Supplementary Fig. 2b, c, Supplementary Tables 2 and 4, and Methods for details). Despite the presence of two other high-score Ci binding sites in other regions close to the *wg* and *wnt6* genes, they did not drive restricted expression to the A compartment of the wing disc (Supplementary Fig. 3a). We next genetically manipulated Hh expression and signalling with the use of the *sd-gal4* driver, which is ubiquitously expressed in the early wing primordia (Fig. 2h), and addressed the impact on the expression of the *wg¹*-lacZ reporter. Interestingly, the expression pattern of this reporter was expanded throughout the A compartment upon ubiquitous expression of Hh and reduced upon ubiquitous overexpression of Ptc (Fig. 2g), which is known to block Hh signalling when overexpressed[25]. Surprisingly, we observed that expression of *wg¹*-lacZ was not abolished upon Ptc overexpression (Fig. 2g, arrows), suggesting that Hh signalling is not an absolute requirement for the expression of Wg in early wing discs. Consistently, only a small proportion of the Ptc-overexpressing adult flies lacked wings and showed duplicated notal structures (Fig. 2k). The presence of vestigial wings in these animals (Fig. 2j, arrows and magnification) is most probably a consequence of reduced expression of the growth-promoting and Hh-regulated morphogen Dpp in late third instar wing discs (reviewed in ref. 26). These results suggest that, early in development, the enhancer present in the *wg¹* region integrates Hh-dependent and -independent inputs to drive the early expression of Wg responsible for wing fate specification.

The mature wing disc contains the primordia of the adult wing and notum, and this subdivision is generated in second instar by the combined activities of three signalling molecules, namely Wg[17], EGFR ligand Vein (Vn, ref. 27), and Unpaired (Upd) ligand, the latter activating the JAK/STAT pathway[28] (Fig. 2c). While the opposing activities of Wg and Vn, which are expressed in the most distal and proximal portions of the wing disc, specify wing and notum fate respectively, the

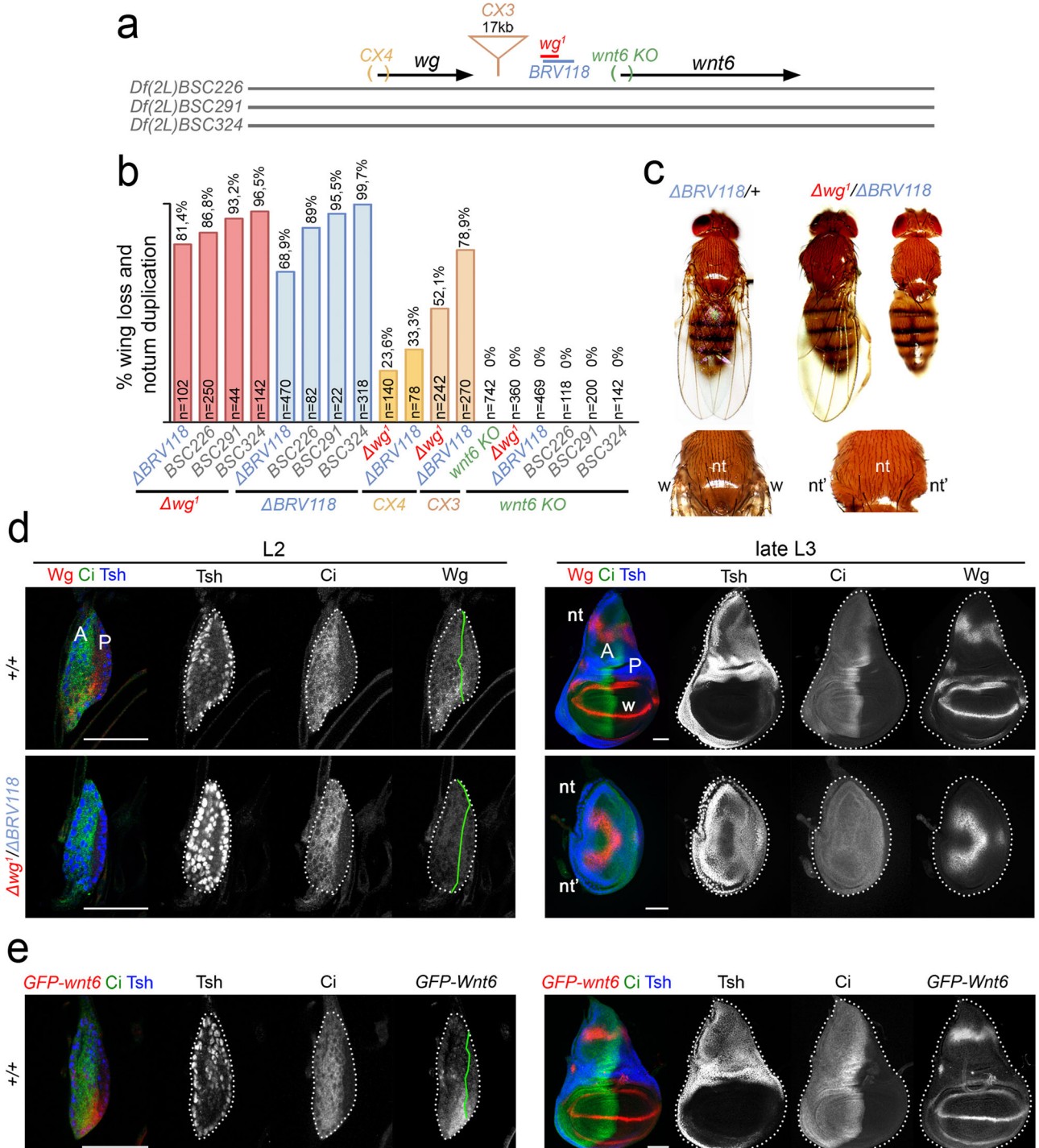

**Fig. 1 | A genomic region required to drive early expression of Wg and wing specification. a** Cartoon depicting the genomic location of the *wg¹*/*BRV118* enhancer between the *wg* and the *wnt6* genes and the deficiencies and mutations affecting these elements. **b** Histogram showing the percentage of wingless flies with duplicated nota of the indicated genotypes. The number of scored heminota are shown. **c** Examples of notum duplications in adult flies of the indicated genotypes. **d**, **e** Second (L2) and late third (L3) instar wing discs

of larvae of the indicated genotypes and stained for Wingless (Wg, red or white, **d**), *GFP-wnt6* transcriptional reporter (red or white, **e**), Ci (green or white, **d**, **e**), and Teashirt (Tsh, blue or white, **d**, **e**). Wing disc contour and the AP boundary are labelled by white and green lines, respectively. Progenitors of wing (w), endogenous notum (nt) and duplicated notum (nt') are marked in (**c**), (**d**). Scale bars, 50 µm. See also Supplementary Fig. 1 and Supplementary Tables 1 and 6.

expression of Upd in the distal portion of the wing disc contributes indirectly to wing fate specification by counteracting the activity of Vn. The observation that *wg¹*-lacZ expression is restricted to the distal portion of the wing disc suggests that it might be negatively regulated by Vn activity in the proximal portion. Consistent with this proposal, expression of a Vn antagonist—a chimeric protein between Vn and the

secreted EGFR antagonist Argos (Vein::Argos, ref. 27)—in the early wing disc gave rise to a proximal expansion in the expression domain of *wg¹*-lacZ (Fig. 2i arrows), and three high-score binding sites for the ETS-family of *Drosophila* transcription factors Yan and Pointed (the most downstream regulator of the EGFR pathway) were identified in the *wg¹* region (red bars in Fig. 2b, see Supplementary Fig. 2b, c,

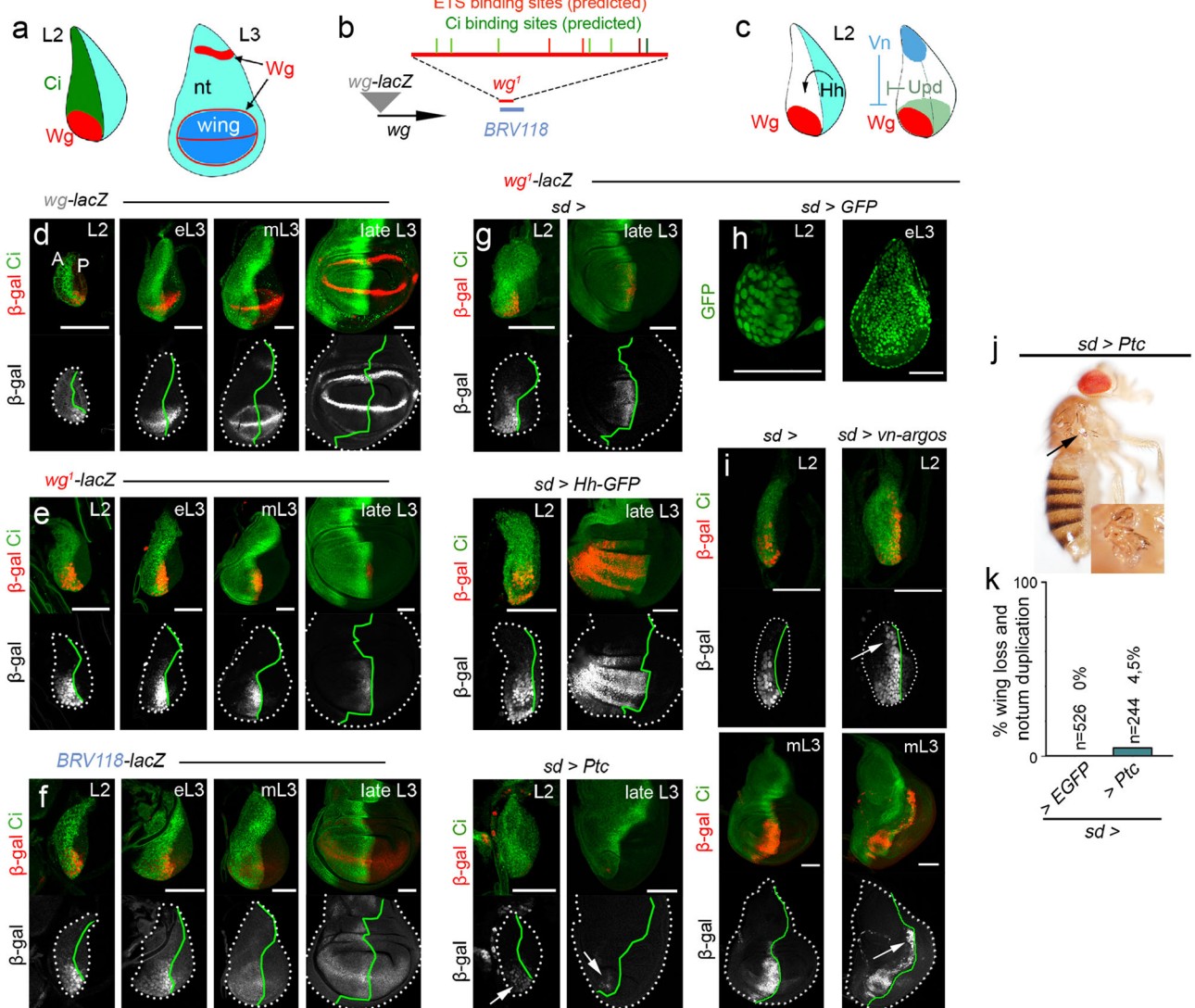

**Fig. 2 | Regulation of the *wg¹*-enhancer by Hh-dependent and -independent mechanisms. a–c** Cartoons depicting the expression of Wg in second (L2) and third (L3) instar wing discs (in **a**), the genomic location of the *wg¹*-enhancer, the Ci (in green) and ETS (in red) binding sites present in it, and the *wg-lacZ* insertion with respect to the *wg* gene (in **b**), and the regulation of *wg* expression in L2 discs by the pre-existing signalling molecules (in **c**). **d–i** Second (L2), early (eL3), mid (mL3) and late third (late L3) instar wing discs of larvae, bearing the *wg-lacZ* enhancer trap (**d**), or the *wg¹-lacZ* (**e, g, i**) and *BRV118-lacZ* (**f**) reporter constructs and stained in (**d–g**) and (**i**) for ß-galactosidase (red or white) and Ci (green) and in h for GFP (green). In

**g–i**, larvae expressed the indicated transgenes under the control of the *sd-gal4* driver. Wing disc contour and the AP boundary are labelled by white and green lines, respectively. Arrows in **g** point to the presence of some lacZ-expressing cells upon Ptc-overexpression and in **i** to the ectopic expression of the lacZ reporters. Scale bars, 50 μm. **j** Adult fly of the indicated genotype showing the most representative phenotype: a vestigial wing (arrow, see also high magnification). **k** Percentages of wingless flies of the indicated genotypes with duplicated nota. Number of scored nota are shown. See also Supplementary Figs. 2 and 3 and Supplementary Tables 1–4 and 6.

Supplementary Tables 3, 4 and Methods for details). These results indicate that the enhancer located in the *wg¹* deletion is regulated in early wing discs by the combined activities of the Hh and Vn signalling molecules.

**The *wg¹*-enhancer is comprised of two *cis*-regulatory modules**
On the basis of sequence conservation with other *Drosophila* species (Supplementary Fig. 2a), we subdivided a 4.7-kb-long region containing the *wg¹* deletion into four fragments [*Alpha* (α), *Beta* (β), *Gamma* (γ) and *Delta* (δ); Fig. 3a] and generated reporter constructs carrying them. The *Alpha* fragment drove expression of *lacZ* to the notum of mature wing discs while *Delta* did not induce the expression of *lacZ* at any time during development (Fig. 3c). The *Gamma* fragment, which contains a high-score Ci binding site (Supplementary Fig. 2b, c), reproduced the expression pattern of the *wg¹-lacZ* reporter in early and late wing discs (Fig. 3c) and was subjected to regulation by Hh

signalling. Thus, the expression pattern of the *Gamma-lacZ* reporter was expanded throughout the A compartment upon ubiquitous expression of Hh and abolished upon ubiquitous overexpression of Ptc (Fig. 3g). The latter observation indicates that Hh signalling is an absolute requirement for the expression of *Gamma* in early wing discs. Further subdivision of *Gamma* into two evolutionarily conserved fragments (Fig. 3b and Supplementary Fig. 2a) allowed us to identify a 590-bp-long fragment containing the Ci binding site with the highest score (dark green bar in Fig. 3a, b) that was sufficient to reproduce the expression pattern of the *wg¹-lacZ* reporter (Fig. 3d). Mutating this Ci binding site compromised, but did not completely abolish, the ability of *Gamma* and the 590-bp-long fragment to drive sustained and restricted expression of *lacZ* to the ventral anterior wedge of early wing discs (Fig. 3d). This observation suggests that a second Ci binding site present in this region but with a lower score (light green bar in Fig. 3a, b, Supplementary Fig. 2b and Supplementary Table 2) is

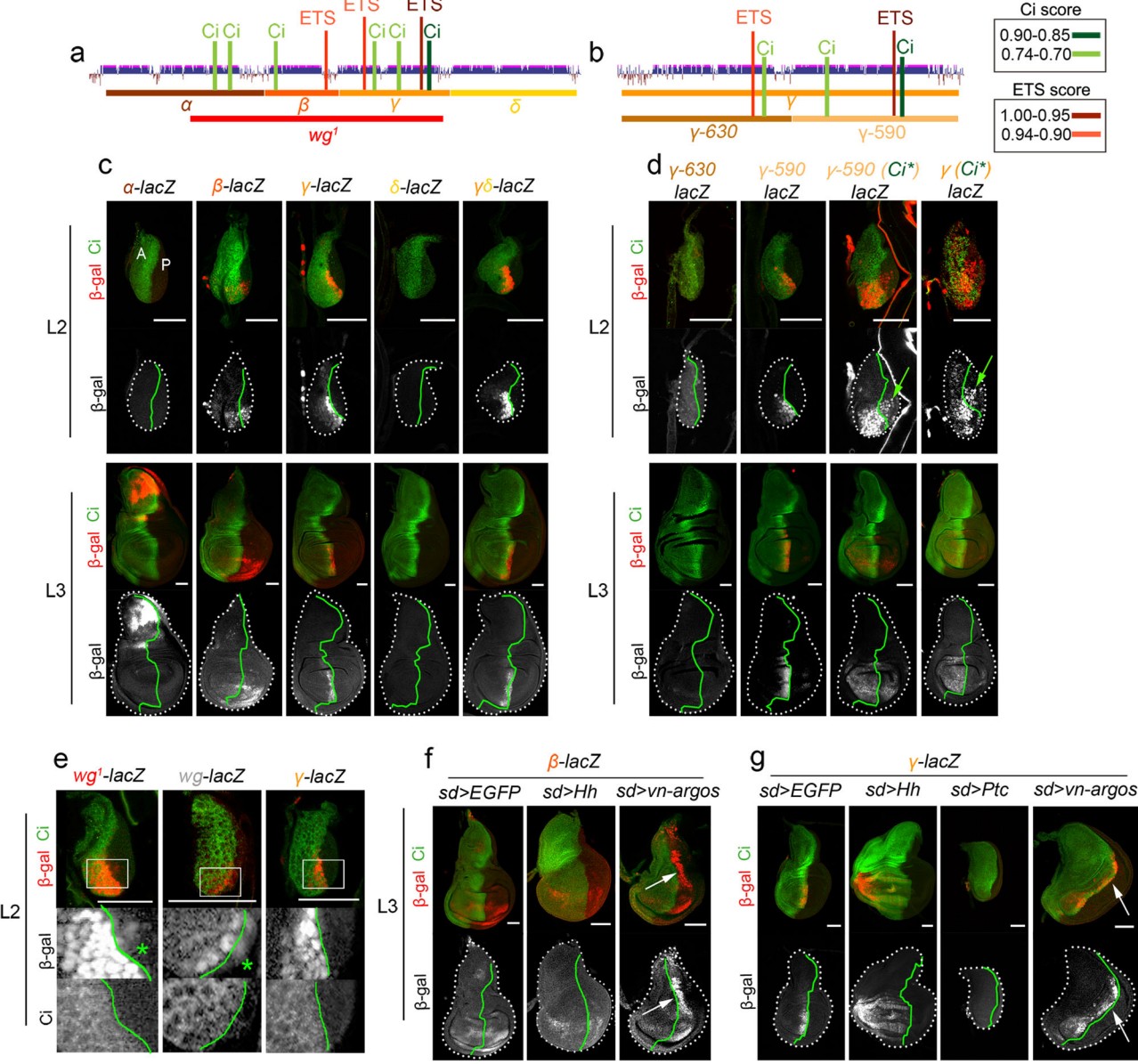

**Fig. 3 | Two independent CRMs within the *wg¹*-enhancer. a, b** Cartoons depicting the evolutionarily conserved modules spanning the *wg¹*-enhancer and the presence of bioinformatically predicted Ci (in green) and ETS (in red) binding sites with the scores shown in the table. **c–g** Second (L2) and late third (L3) instar wing imaginal discs of larvae bearing the indicated reporters and stained for ß-galactosidase (red or white) and Ci (green). Green arrows in (**d**) and green stars in (**e**) mark lacZ-expressing cells in the P compartment. In **f, g**, larvae also expressed the indicated transgenes under the control of the *sd-gal4* driver, and arrows point to the ectopic expression of the lacZ reporters. Wing disc contour and the AP boundary are labelled by white and green lines, respectively. Scale bars, 50 µm. See also Supplementary Fig. 2 and Supplementary Tables 1–4.

functionally relevant in regulating Wg expression. Despite the presence of a predicted low-score Ci binding site in the 630-bp-long fragment (light green bar in Fig. 3a, b, and Supplementary Table 2), this fragment did not drive expression in this context (Fig. 3d), thereby suggesting that this site is not functional in the wing disc. As with the *wg¹*-lacZ reporter, expression of *Gamma-lacZ* was restricted to the distal portion of the wing (Fig. 3c), suggesting that this CRM responds to the activity of Vn. Consistently, expression of Vn::Argos in the early wing disc gave rise to proximal expansion in the expression of *Gamma-lacZ* (Fig. 3g, arrows), and two high-score binding sites for the ETS transcription factors Yan and Pointed were identified in this CRM (red bars in Fig. 3a, b and Supplementary Tables 3 and 4). These results indicate that *Gamma* is a CRM that integrates the positive input of Hh and negative input of Vn to drive expression of Wg to a ventral anterior wedge in the early wing disc.

The *Beta* fragment drove expression of *lacZ* to the most distal region of early and late wing discs (Fig. 3c), thereby suggesting that the Vn-dependent input into the *wg¹*-lacZ reporter might be also mediated by this CRM. Indeed, expression of Vn::Argos in the early wing disc gave rise to a proximal expansion in the expression pattern driven by *Beta* (Fig. 3f), and one high-score binding site for Yan and Pointed was identified in this CRM (red bars in Fig. 3a, b and Supplementary Tables 3 and 4). Despite the presence of a predicted low-score Ci binding site in *Beta* (light green bar in Fig. 3a, b, Supplementary Fig. 2b and Supplementary Table 2), this fragment drove expression of *lacZ* to both A and P cells in wild-type larvae (Fig. 3c), and this expression was not expanded throughout the A compartment upon ubiquitous expression of Hh (Fig. 3c), thereby indicating that this binding site is not functionally relevant in this context. All these results show that the activities of the *Gamma* and *Beta* CRMs are restricted to the distal part

of the wing disc by Vn to drive Wg expression and wing fate specification in the distal part of the wing primordium, and that *Gamma* and *Beta* CRMs integrate Hh-dependent and -independent inputs, respectively. Consistently, we observed that *wg¹*-lacZ (containing the two CRMs) and the *wg-lacZ* enhancer trap (responding to the two CRMs) drove expression of *lacZ* not only to A but also to P cells (Fig. 3e), and that expression of *Gamma-lacZ* was restricted to the A compartment (Fig. 3e). We noticed that expression of *lacZ* driven by *Beta* and *Gamma* CRMs in second instar wing discs persisted in later stages even in the presence of *Delta* (Fig. 3c), a genomic region that has been shown to block the activity of this enhancer upon tissue damage in mature wing discs[13]. These results suggest that the regulatory elements responsible for turning *Gamma* and *Beta* CRMs off during normal development are not present in these genomic regions.

### The two CRMs act redundantly to drive wing specification

The above results also suggest that Wg expression in the early wing disc is regulated in a redundant manner by two independent CRMs: *Gamma*, which responds to the combined activities of Hh and Vn; and *Beta*, which responds only to the negative input of Vn. To experimentally confirm the biological relevance of this proposal and the functional redundancy of the two CRMs in driving Wg expression and wing fate specification, we used the CRISPR/Cas9 technique to generate targeted deletions within the *Gamma* and *Beta* CRMs (Fig. 4a, b). Targeted mutation of the highest-score Ci binding site in *Gamma* or deletion of the 590-bp-long fragment (*Δγ-590*) did not cause a wingless phenotype in adults when homozygous, and only 0.39% of flies lacking the whole *Gamma* CRM showed a wingless phenotype (Fig. 4c). Similarly, deletion of *Beta*, which drove expression in the distal side of the early wing disc, did not cause any overt phenotype in adults (Fig. 4c). It was necessary to delete both *Beta* and *Gamma* to induce a wingless phenotype in adults (Fig. 4c, d). Interestingly, the penetrance of the wingless phenotype was up to 80% upon deletion of both *Beta* and *Gamma* (in *Δβγ* homozygous or *Δβγ/ΔBRV118* transheterozygous flies, Fig. 4c). As expected, the early expression of Wg was lost in *Δβγ* homozygous wing discs, thus causing mature wing discs to lack wing progenitors and duplicate the notum primordium (Fig. 4d, e). The deletion of *Gamma* over larger deletions containing the *wg¹*-enhancer (in *Δγ/Δβγ* or *Δγ/ΔBRV118* flies) induced a wingless adult phenotype, although at a lower penetrance (12% in *Δγ/Δβγ* flies and 37% in *Δγ/ΔBRV118* flies, Fig. 4c). Even deletion of the 590-bp-long fragment over *ΔBRV118* was able to cause this phenotype, but in this case penetrance was extremely low (1.65%, Fig. 4c). In contrast, deletions of *Beta* did not cause a reproducible wingless adult phenotype over the same deletions (Fig. 4c). Taken together, these results demonstrate the functional redundancy of *Gamma* and *Beta* CRMs in driving the expression of Wg in early wing discs and in specifying the wing. We observed that the penetrance of the wingless adult phenotype was higher in *Δγ* than in *Δγ-590* transheterozygous (Fig. 4c) pointing to a potential role of the remaining 630-bp-long fragment in the regulation of *wingless* expression despite its inability to drive *lacZ* expression (Fig. 3d). Whether this fragment contributes to transcription factor accessibility, enhancer-promoter interactions or chromatin conformation remains to be further investigated.

Adult flies mutant for the original *wg¹* deletion, which covers the *Gamma* and *Beta* CRMs and part of the *Alpha* region, presented a small effect on the shape of the eyes (Supplementary Fig. 4a, see also ref. 29). Also, some *wg¹* mutant individuals lost notum structures in the adult and beared very small wing and haltere imaginal discs (Supplementary Fig. 4b, c, see also ref. 15). Interestingly, none of these phenotypes occurred in *Δβγ* homozygous flies (Fig. 4d and Supplementary Fig. 4a, c). The phenotypes observed in *wg¹* individuals are consistent with the fact that the *wg¹-lacZ* reporter is expressed in other parts of the developing fly, including the eye-imaginal primordium and the larval notum (Supplementary Fig. 2b).

Interestingly, the expression pattern of the eye-imaginal primordium was also reproduced by the *Alpha* and *Beta* regions, and *Alpha* drove expression also to the presumptive notum in the developing wing (Supplementary Fig. 4d–f). Although *Alpha* and *Gamma* drove expression of the reporter in the leg discs (Supplementary Fig. 4d, f), no overt phenotype was observed in the adult appendage of *wg¹* mutant flies (not shown). These results point to the presence of three partially overlapping functional enhancers in this region (Supplementary Fig. 4g) and unravel a 1.8-kb-long wing-specific enhancer comprised of the *Beta* and *Gamma* CRMs.

### The two CRMs respond to JNK to drive wing regeneration

Later in development and once the wing primordium has been specified, Wg, emerging from a central stripe that corresponds to the future wing margin, acts as mitogenic molecule and drives wing growth[18,19]. Wnt6, which is also expressed in the future wing margin (Fig. 1e), but is less efficient at activating the canonical pathway than Wg[30], also contributes to wing growth[18]. Tissue injury induces the expression of these two ligands as a consequence of JNK signalling acting on the *wg¹*-enhancer, which is populated by five bioinformatically predicted high-score AP1 binding sites (blue bars in Fig. 5a, Supplementary Fig. 2b, c and Supplementary Table 5), and Wg has been shown to play a functional role in driving compensatory proliferation and regeneration of the developing wing[4,13,31]. However, experimental settings to genetically induce ablation of the wing primordium by different means came up with opposite observations and conclusions on the role of Wg in wing regeneration[32–34]. In some cases, functional experiments to deplete Wg expression and address its role in wing regeneration upon injury were also unable to completely circumvent the developmental requirements of Wg in wing fate specification and wing growth. No attempt to analyse the specific contribution of Wnt6 to wing regeneration was reported either. Thus, using our collection of expression reporters and CRISPR/Cas9-targeted deletions, we decided to revisit the role of Wg, Wnt6, the *wg¹*-enhancer and the newly identified CRMs in regeneration.

Two distinct experimental settings were used to address whether Wg is required for tissue regeneration upon ablation. Overexpression of the *Drosophila* TNF-α homologue Eiger was used in refs. 4,13, to activate the JNK pathway and induce the apoptosis of most wing pouch cells. Overexpression of the pro-apoptotic gene *reaper* was used in ref. 32 to directly induce apoptosis. In both cases, transgene expression was driven by the *rotund-gal4* (*rn-gal4*) driver, which is expressed in those cells that will give rise to the adult wing (Fig. 5b, region coloured in yellow). The GAL4/UAS system was combined with the temperature-sensitive version of the Gal4 repressor Gal80 (Gal80ts) to drive transgene expression during a discrete period of time in early third instar wing discs and to analyse the capacity of the remaining tissue to give rise to a normal adult wing as a consequence of compensatory proliferation. We used two different transactivation systems (Gal4/UAS and LexA/LexO) and shortened the induction period to 11 h (in the case of Reaper) or 16 h (in the case of Eiger) in early third instar larvae (Fig. 5b). We used *rn-gal4* to drive Eiger expression and *spalt-lexA*, which is expressed in a central region of the presumptive wing (Fig. 5b, region coloured in brown), to drive *reaper* expression. We first compared the ability of these two transgenes to drive ectopic activation of *wg¹*-lacZ and expression of the Wg and Wnt6 genes. Both transgenes triggered robust activation of *wg¹*-lacZ, and ectopic expression of an endogenous Wg-GFP fusion protein, the *wnt6* gene, and MMP1, a bona fide target gene of JNK in *Drosophila*[35] (Fig. 5c–g). All these reporters were expressed mainly in apoptotic cells, as visualised by their pyknotic nuclei.

The ability of the *wg¹*-enhancer to drive lacZ expression upon Eiger expression was drastically compromised when mutating four out of the five existing AP1 binding sites (Fig. 6b, mutated AP1 binding sites are marked with stars in Fig. 6a, see 'Methods' for details). We observed that the developmental expression of the *wg¹*-enhancer was largely

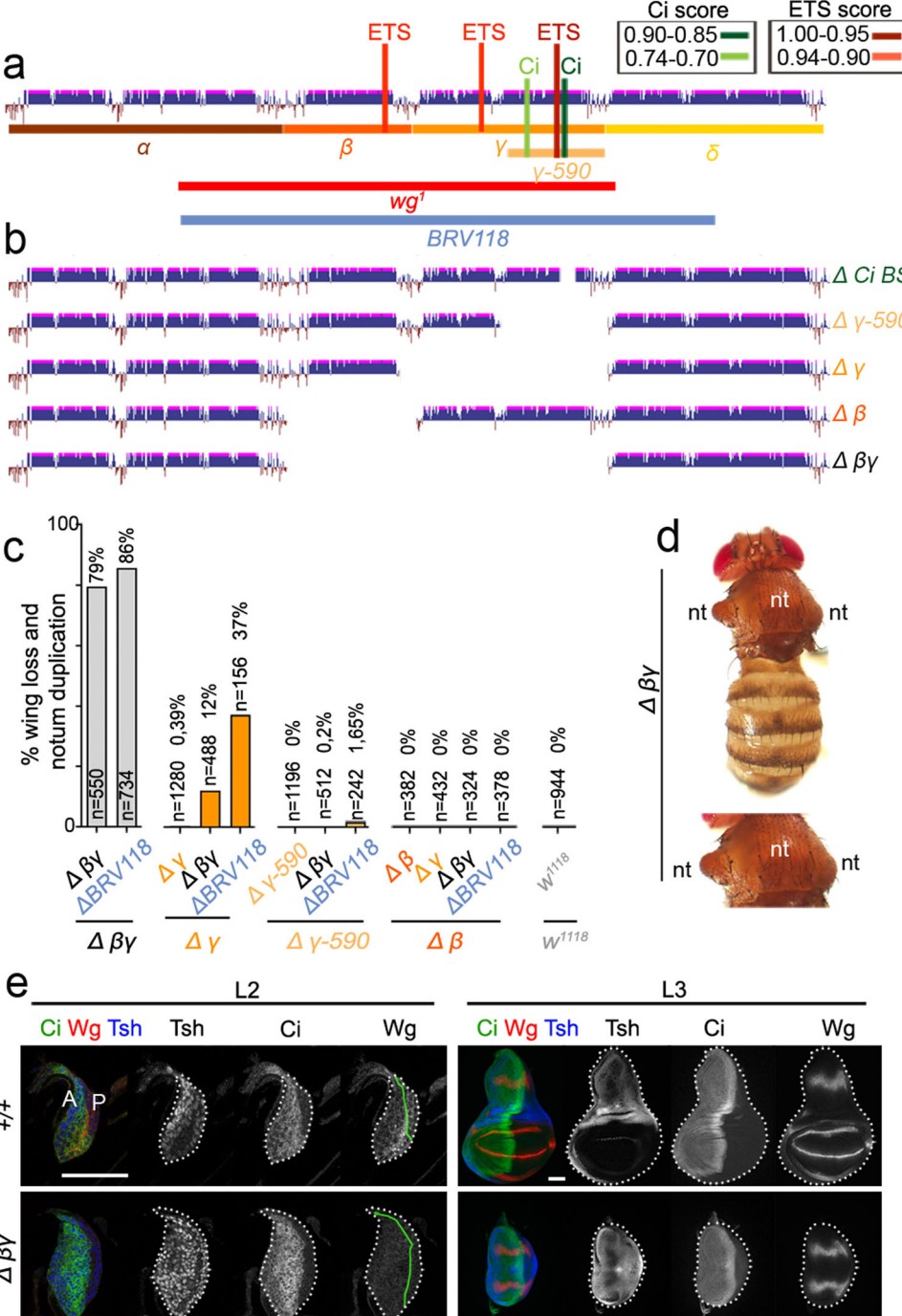

**Fig. 4 | Functional redundancy within the *wg¹*-enhancer to drive wing fate specification. a**, **b** Cartoons depicting in **a** the evolutionarily conserved modules spanning the *wg¹*-enhancer and the presence of bioinformatically predict predicted Ci (in green) and ETS (in red) binding sites with the scores shown in the table, and in **b** the CRISPR/Cas9-induced deletions of the indicated elements. **c** Histogram plotting the percentage of wing loss and notum duplication in flies of the indicated genotypes. The number of scored heminota is also shown. **d** Examples of notum duplication of an adult fly of the indicated genotype. **e** Second (L2) and late third (L3) instar wing discs of larvae of the indicated genotypes and stained for Wg (red or white), Ci (green or white), and Teashirt (Tsh, blue or white). Wing disc contour and the AP boundary are labelled by white and green lines, respectively. Scale bars, 50 μm. See also Supplementary Figs. 2 and 4 and Supplementary Tables 1–4 and 6.

unaffected by these mutations (Supplementary Fig. 4e). We next used our *lacZ*-reporter constructs carrying the evolutionarily conserved *Alpha*, *Beta*, *Gamma* and *Delta* genomic regions to further validate the bioinformatically predicted AP1 binding sites, and targeted deletions of these fragments to functionally address their contribution to JNK-driven expression of Wg and Wnt6 and to wing regeneration. Our results indicate that only *Beta* and *Gamma* are induced in wing discs expressing Eiger, and that JNK has an impact on both the 630-bp- and

590-bp-long fragments comprising the *Gamma* CRM (Fig. 6b). The levels of Wg expression induced by Eiger were significantly reduced in *Δβγ* homozygous flies and partially reduced in flies where either the *Gamma* or *Beta* CRMs were deleted (Fig. 6c, d). These findings thus point to the functionally independent role of these CRMs in mediating JNK-driven Wg expression. Interestingly, Eiger was able to cause ectopic expression of Wg even upon deletion of these two CRMs, thus indicating the presence of other genomic regions acting on Wg and

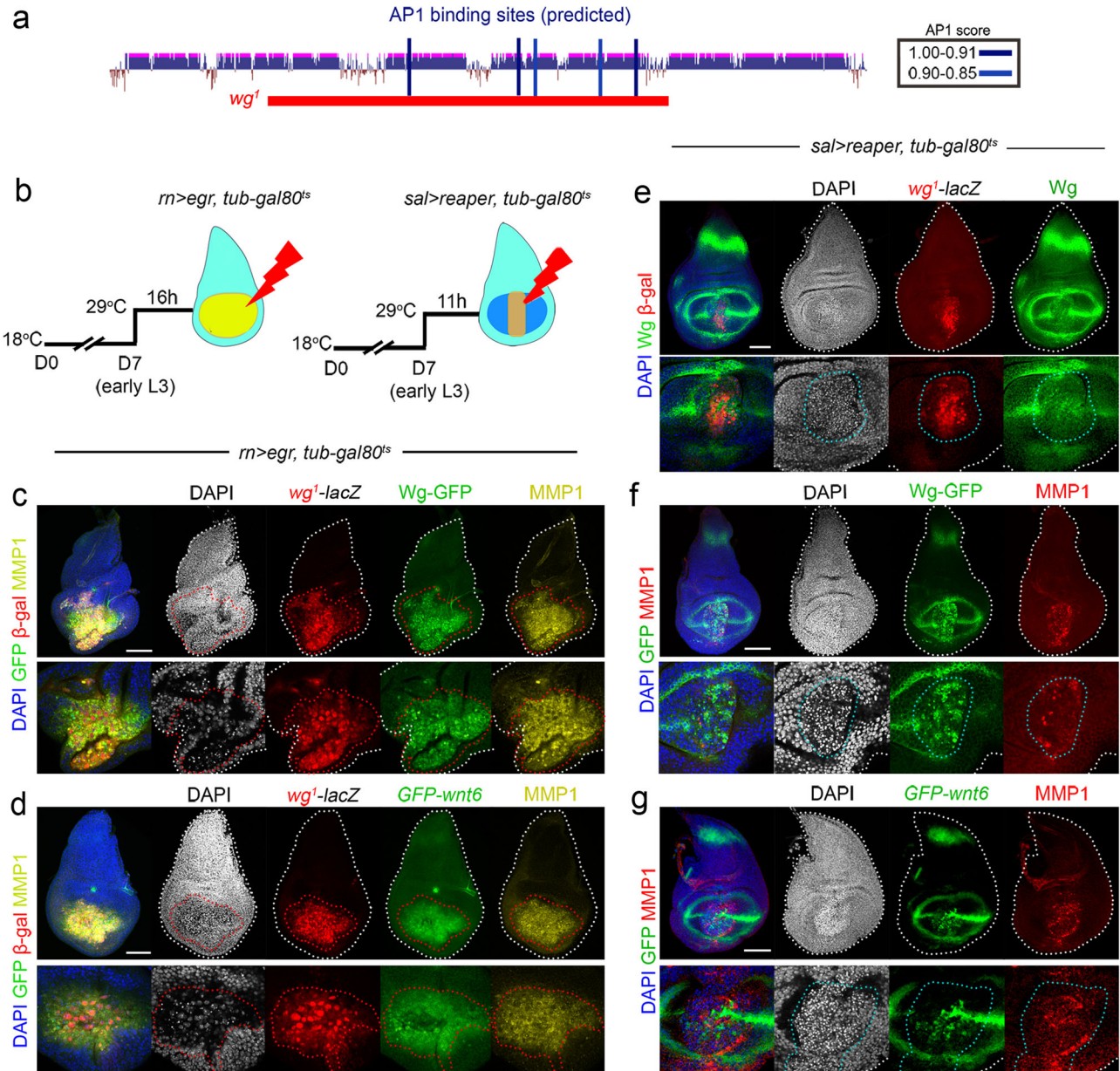

**Fig. 5 | The *wg¹*-enhancer is activated in apoptotic cells upon tissue injury.**
**a** Cartoon depicting the presence of bioinformatically predicted high-score AP1 binding sites (in blue) in the *wg¹*-enhancer with the scores shown in the table. **b** Schematic representation of the Eiger- and Reaper-dependent wing ablation systems. Larvae were raised at 18 °C for 7 days (D7) and switched to 29 °C for 16 h (**c**, **d**) or 11 h (**e**–**g**) to visualise gene expression. **c**–**g** Third instar wing discs of larvae bearing the indicated reporters, after Eiger (**c**, **d**) or Reaper (**e**–**g**) expression and stained for *wg¹-lacZ* expression (antibody to ß-galactosidase in red, **c**, **d**, **e**), Wg-GFP (green, **c**, **f**), *GFP-Wnt6* (green, **d**, **g**), Wg (green, **e**), MMP1 (yellow in **c**, **d**, and red in **f**, **g**), and DAPI (blue or white). Wing disc contours are labelled by a white line, cells expressing the *wg¹*-enhancer by a red line (**c**, **d**) and apoptotic cells by a blue line (**e**–**g**). Higher magnification of the wing pouch is shown in lower panels. Scale bars, 50 μm. See also Supplementary Fig. 2 and Supplementary Tables 1 and 5.

responding to JNK. The presence of Gal80[ts] also allowed us to express Eiger in the wing pouch for a short period (16 h, Fig. 6e), thus killing a good fraction of developing wing cells, and to address the regenerative capacity of the remaining tissue in flies where the *Gamma* or *Beta* CRMs had been deleted. Deletion of either CRM in homozygosis did not cause any wing phenotype by themselves (Fig. 4c) but significantly reduced the regenerative capacity of the wing (Fig. 6f, g). This capacity was even more compromised when both CRMs were deleted in homozygosis (Fig. 6f, g). In this case, those flies presenting a notum duplication and absence of wing tissue, as a result of the developmental role of both CRMs in wing fate specification, were not scored.

Wg is a potent mitogenic molecule in wing pouch cells[18,19] and Wnt6 contributes to positively regulate this mitogenic activity[18].

Consistently, Wg expression levels and mitotic activity in wing discs subjected to Eiger expression, which is increased when compared to undamaged discs[4], were reduced in *Δγ*, *Δβ* and *Δβγ* homozygous individuals (Fig. 6c, d). Similar results were obtained when monitoring cells in S-phase (Fig. 6c, d). Not only Wg but also Wnt6 contributed to the regenerative capacity of the wing, as RNAi-mediated depletion of Wnt6 compromised this capacity and reduced the mitotic activity observed in regenerating wing primordia (Fig. 6h, i). All these results indicate that the two evolutionarily conserved CRMs of the wing-specific enhancer respond to JNK and contribute, in a functionally redundant manner, to tissue regeneration, thus reinforcing the role of Wg as a mitogen and Wnt6 as a positive modulator[18].

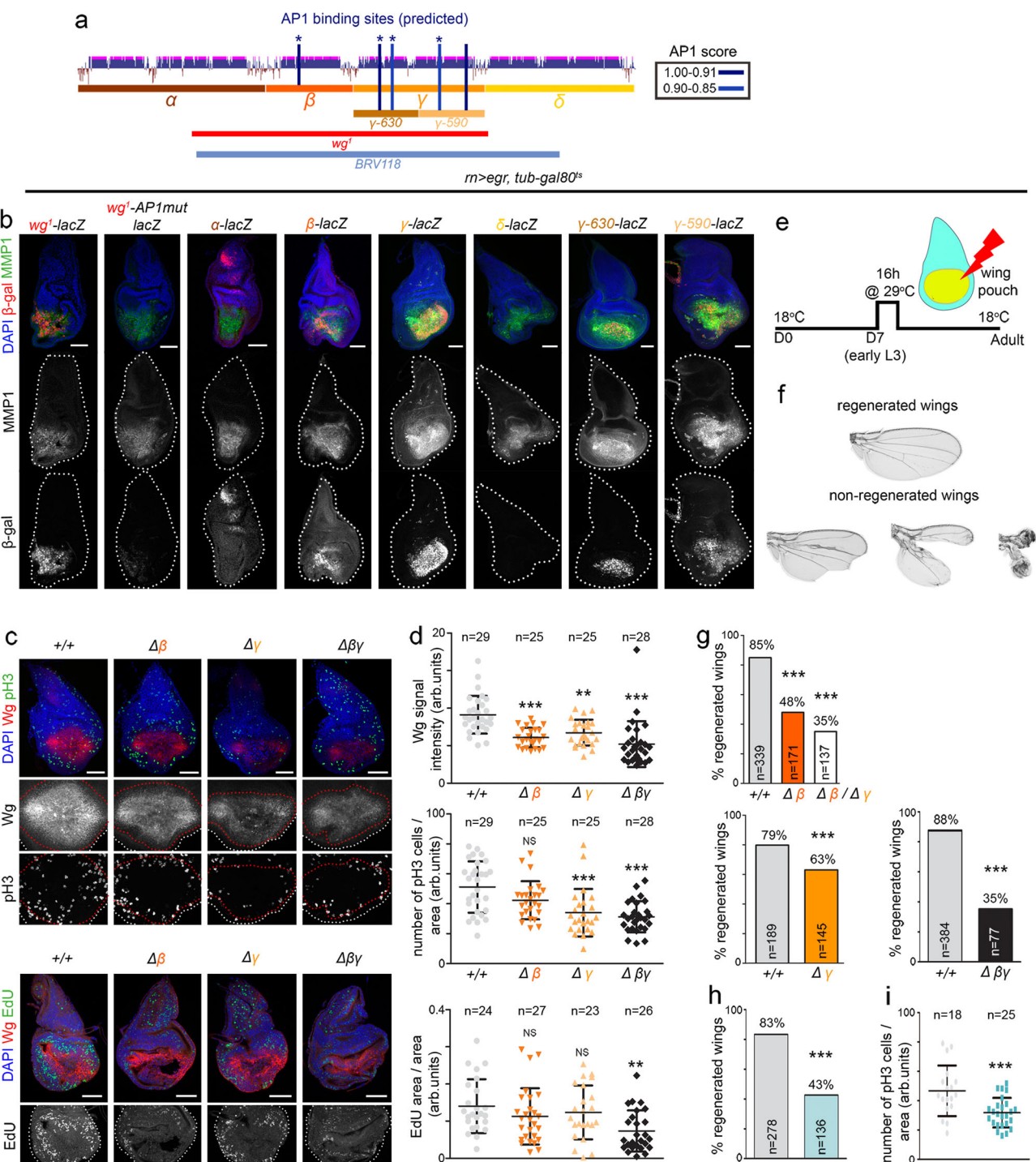

### The two CRMs respond to JNK to drive malignant overgrowth

Vertebrate and invertebrate tissues with strong regenerative capacity, such as the mammalian liver and fly wing primordium, use almost the same signalling molecules to regenerate a missing part or to induce tumorigenesis in situations in which deleterious cells have not been removed by apoptosis. In the case of the developing fly wing, JNK-driven expression of Wg and Wnt6 induces compensatory proliferation and wing regeneration upon tissue injury (above results and ref. 13), and JNK-driven expression of Wg in tissues subjected to chromosomal instability (CIN)−a high rate in the gain or loss of chromosomes during mitosis and a hallmark or most solid tumours in humans− contributes to the production of tumour-like overgrowths upon additional blockage of the apoptotic pathway[36,37]. We decided to

analyse whether the two ligands and the wing-specific enhancer were activated in CIN tissues, and whether the two CRMs also contributed to the resulting tumour-like overgrowths in a functionally redundant manner. Interestingly, not only Wg but also Wnt6 was ectopically induced in CIN tissues (Fig. 7a), *wg¹*-lacZ, *Beta* and *Gamma* CRMs, and the 630-bp- and 590-bp-long fragments comprising the *Gamma* CRM were activated (Fig. 7b) and their expression levels were drastically reduced by expressing a dominant negative version of JNK (Bsk in *Drosophila*, Fig. 7c). Mutating four out of the five existing AP1 binding sites in the *wg¹*-enhancer (labelled with a star in Fig. 6a) compromised its ability to drive *lacZ* expression in CIN tissues (Fig. 7b). Deletion of either *Beta* or *Gamma* in homozygosis significantly reduced the CIN-induced tissue overgrowth, as well as the levels of Wg expression

**Fig. 6 | Functionally redundant CRMs in the *wg¹*-enhancer respond to JNK and contribute to wing regeneration. a** Cartoon depicting the evolutionarily conserved modules spanning the *wg¹*-enhancer and the presence of bioinformatically predicted AP1 binding sites (in blue) with the scores shown in the table. Stars mark mutated AP1 binding sites in the *wg¹-AP1mut-lacZ* reporter. **b, c** Third instar wing discs of larvae, subjected to the expression of Eiger under the control of the *rn-gal4* driver for 16 h, bearing the indicated reporters in **b**, and stained for ß-galactosidase (red or white, **b**), MMP1 (green or white, **b**), Wg (red or white, **c**), pH3 (green or white, **c** top panels), EdU (green or white, **c** bottom panels), and DAPI (blue). Wing disc contours are labelled by white lines in (**b**). Scale bars, 50 μm. **d, g, h, i** Scattered plots (**d, i**) representing Wg signal intensity (in arbitrary units, **d**), number of pH3-positive cells per area (in arbitrary units, **d, i**), and EdU incorporation per area (in arbitrary units, **d**), and histograms (**g, h**) plotting the percentages of fully regenerated wings of individuals of the indicated genotypes. The number of scored wing

discs or wings is shown in (**d, g, h, i**), and the area where mitotic activity was quantified in (**d, h, i**) is labelled by a red line in (**c**). Mean and SD (**d, i**) are shown. All statistical tests (Anova in **d, i**, logistic regression/Wald test statistic in **g, h**) are two-tailed and, in case of more than two conditions, Dunnett's multiple comparison correction against a common control was performed. Statistically significant differences are shown: NS, $p > 0.05$; **$p < 0.01$; ***$p < 0.001$. *p* values **d:** Wg = 0.00002 ($\Delta\beta/\Delta\beta$), 0.0016 ($\Delta\gamma/\Delta\gamma$), $4.5 \times 10^{-11}$ ($\Delta\beta\gamma/\Delta\beta\gamma$); pH3 = 0.2 ($\Delta\beta/\Delta\beta$), 0.00007 ($\Delta\gamma/\Delta\gamma$), 0.000002 ($\Delta\beta\gamma/\Delta\beta\gamma$), EdU = 0.2 ($\Delta\beta/\Delta\beta$), 0.53 ($\Delta\gamma/\Delta\gamma$), 0.001 ($\Delta\beta\gamma/\Delta\beta\gamma$), **i:** pH3 = 0.0006. **e** Schematic representation of the Eiger-dependent wing ablation and regeneration system. Larvae were raised at 18 °C for 7 days (D7), switched to 29 °C for 16 h and allowed to develop at 18 °C until adulthood. **f** Examples of the resulting fully regenerated and non-regenerated adult wings. See also Supplementary Fig. 2, Supplementary Tables 1, 5 and 7. Source data are provided as Source data file.

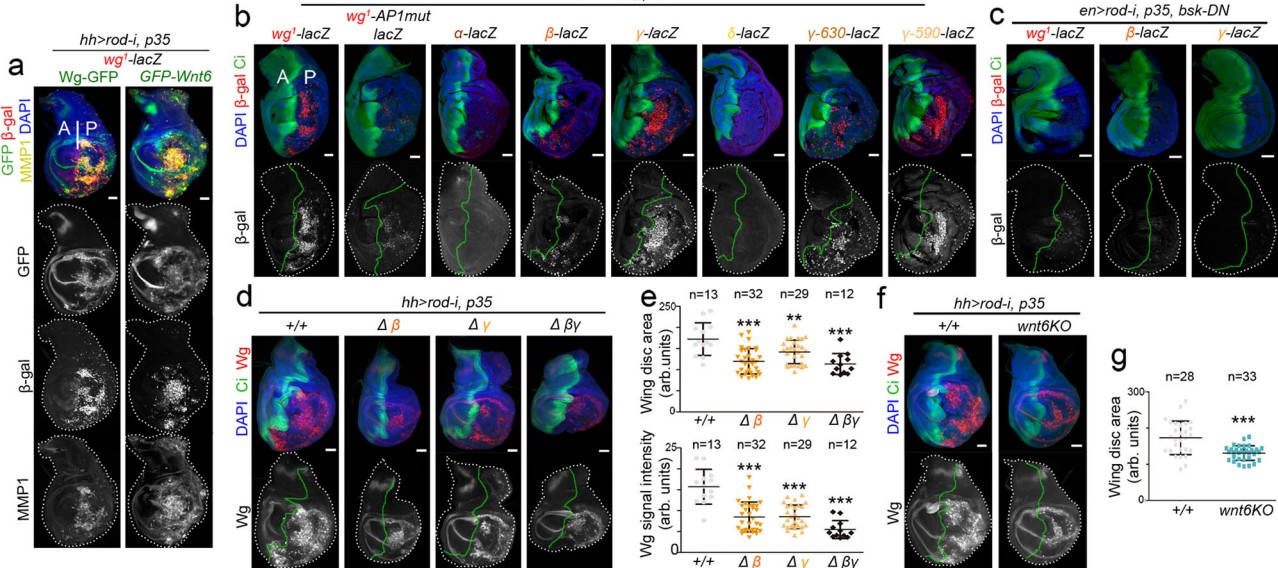

**Fig. 7 | Functionally redundant CRMs in the *wg¹*-enhancer respond to JNK and contribute to the growth of Chromosomal Instability-induced tumours.** **a–d, f** Third instar wing discs of larvae bearing the indicated reporters, subjected to the expression of the indicated transgenes under the control of the *en-gal4* (**b, c**) or *hh-gal4* (**a, d, f**) drivers for four days and stained GFP (**a**, green or white), ß-galactosidase (**a, b, c**, red or white), MMP1 (**a**, yellow or white), Wg (**d, f**, red or white), Ci (**b, c, d, f**, green), and DAPI (blue). Gal4 drivers are expressed in the posterior (P) compartment and Ci is used to label the anterior (A) compartment. Wing disc contours are labelled by white lines and the AP boundary by a green line.

Scale bars, 50 μm. **e, g** Scattered plots representing wing disc area and Wg signal intensity (in arbitrary units) of the indicated genotypes. Mean and SD are shown. Number of wing discs is shown. All statistical tests are two-tailed Anova tests and, in case of more than two conditions, Dunnett's multiple comparison correction against a common control was performed. Statistically significant differences are shown: NS, $p > 0.05$; **$p < 0.01$; ***$p < 0.001$. *p* values **e:** Wing disc area = $4.2 \times 10^{-6}$ ($\Delta\beta/\Delta\beta$), 0.009 ($\Delta\gamma/\Delta\gamma$), 0.00001 ($\Delta\beta\gamma/\Delta\beta\gamma$); Wg = $1.1 \times 10^{-7}$ ($\Delta\beta/\Delta\beta$), $7.4 \times 10^{-6}$ ($\Delta\gamma/\Delta\gamma$), $2.5 \times 10^{-12}$ ($\Delta\beta\gamma/\Delta\beta\gamma$), **g:** Wing disc area = 0.000018. See also Supplementary Table 7. Source data are provided as Source data file.

(Fig. 7d–f). The effects on tissue overgrowth and Wg expression were even stronger when both fragments were deleted simultaneously (in *Δβγ* larvae, Fig. 7d–e, we obviously selected those mutant wing discs bearing a wing primordium). Not only Wg[36] but also Wnt6 contributed to the CIN-induced tissue overgrowth, as monitored by the effects on tissue size of a null allele of *wnt6* (Fig. 7f, g). All these results indicate that the two evolutionarily conserved CRMs of the wing-specific enhancer respond to JNK and contribute, in a functionally redundant manner, to CIN-induced tissue overgrowth by driving the expression of both Wg and Wnt6.

## Discussion

### A *wingless* enhancer devoted to wing specification

Since the serendipitous discovery of *wg¹* 50 years ago[9], one of the most remarkable aspects of this mutation is the restriction of its phenotype to the absence of wings, hence its name *wingless* and that of the whole family of Wnts (a fusion of *wingless* and the vertebrate homologue, *integrated* or *int-1*). After five decades of research, this wing-specific

phenotype contrasts with the highly pleiotropic effects of the *wingless* gene during the embryonic and larval development of the fly. Here we have solved this conundrum by presenting experimental evidence that the *wg¹* phenotype is due to the loss of an enhancer whose functional requirement is restricted to the developing wing primordium. We combined CRISPR/Cas9-mediated deletions and reporter assays to identify and narrow down the wing-specific enhancer to a 1.8-kb-long genomic region that drives Wg expression and wing fate specification in early wing discs. This enhancer contains two highly evolutionarily conserved CRMs, *Beta* and *Gamma*, which are activated in a redundant manner by a combination of pre-existing signalling molecules (Fig. 8a). Whereas Vn/EGFR signalling emanating from the most proximal part of the wing disc restricts the expression of *Beta* and *Gamma* to the most distal portion of the wing, Hh emanating from P cells induces the expression of *Gamma* in A cells (Fig. 8a). The redundant use of pre-existing signals acting independently on these two evolutionarily conserved CRMs to trigger Wg expression and wing fate specification reveals a highly robust mechanism to ensure wing development.

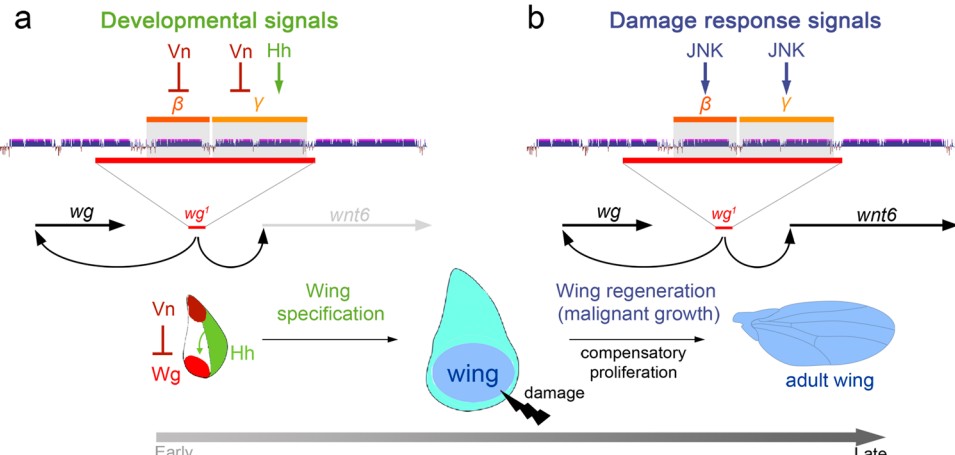

**Fig. 8 | A Wnt enhancer devoted to wing fate specification, regeneration and malignant growth. a, b** Cartoons depicting the redundant roles of Beta and Gamma CRMs of the *wg¹*-enhancer in responding to developmental (**a**) or damage response signals (**b**), and in driving wing specification through the activity of Wingless (**a**), and wing regeneration or malignant growth through the activities of Wingless and Wnt6 (**b**). Wnt6 has no major role in wing specification despite the fact that it is expressed during development in the same expression pattern as Wg.

Identification of a wing-specific enhancer driving Wg expression and wing fate specification from body wall cells in the early larval wing disc reinforces the proposal that insect wings evolved in evolution as an extension of the dorsal thorax and not directly from a proximal leg component[38,39].

## Functional redundancy in driving wing regeneration

Wnts can act not only as cell fate determinants but also as mitogenic molecules. In the developing wing primordium of *Drosophila*, these two activities are separated in time and take place in two consecutive developmental stages. Whereas Wg drives wing fate specification at early larval stages[17], it promotes wing growth at later stages by mediating the organising activity of the signalling centre located in the developing wing margin[18,20]. Research carried out in the wing primordium pointed to a role of Wg in compensatory proliferation upon tissue injury[4,31], but this proposal was subsequently disputed[32,34]. The identification of the wing-specific enhancer as the one responding to the activation of the stress-induced signalling pathway JNK upon several types of tissue injuries, such as Eiger induction, physical damage or ionizing radiation-induced DNA damage, contributed to reinforcing the mitogenic role of Wg in injured tissues[13]. Unfortunately, though, functional experiments to deplete Wg expression and address its role in wing regeneration upon injury did not fully circumvent the developmental requirements of Wg in wing fate specification and growth[13,32]. Here we combined the use of CRISPR/Cas9-mediated deletions, which have no major effects on wing fate specification or growth, and reporter assays to demonstrate that the two highly evolutionarily conserved CRMs, *Beta* and *Gamma*, are activated by JNK and are functionally required in a redundant manner to drive Wg expression and compensatory proliferation upon tissue damage (Fig. 8b). The redundant and shared use of the same regulatory elements in wing fate specification and regeneration unravels a highly evolutionarily robust mechanism to ensure the development of the wing, probably the most important evolutionary innovation in insects, not only during normal development but also under stress conditions.

## Wing regeneration and tumorigenesis: commonalities

Vertebrate and invertebrate tissues with strong regeneration capacity, such as the *Drosophila* wing primordium, use almost the same signalling molecules to regenerate a missing part or to induce tumorigenesis when harmful cells are maintained in the tissue upon apoptosis inhibition. In the last few years, our lab has underscored the deleterious consequences of maintaining CIN- or DNA damage-induced highly aneuploid cells in the wing epithelium[36]. CIN or DNA damage causes tumour-like overgrowths upon blockade of the apoptotic machinery, and this tumorigenic response relies mainly on the production of highly aneuploid cells that delaminate from the epithelium and activate a JNK-dependent transcriptional response. Here we present evidence that the wing-specific enhancer of the *wg* gene is activated in the same types of cells in regenerating and tumorigenic tissues. Whereas it is activated in dying cells within regenerating tissues, this activation takes place in highly deleterious aneuploid cells within CIN-tissues expressing the apoptosis inhibitor p35. JNK-driven activation of the enhancer in both cases is shown to promote Wg expression and the proliferation and growth of the wing epithelium (Fig. 8b). These results reinforce the proposal that the same type of cells (apoptotic or deleterious cells), the same molecular actor (JNK), and the same molecules (Wingless and Wnt6) mediate both tissue regeneration and tumorigenesis. The main difference between the two cases is the amount of time deleterious cells spend in the tissue before being removed by immune cells. Interestingly, similar phenomena take place in vertebrate epithelia and exactly the same actors appear to be in place[40].

## A role of Wnt6 in proliferative growth

The fly wing primordium has served as a useful model system to visualise how Wg acts as fate determinant and mitogenic molecule in two consecutive developmental time points. Here we present evidence that Wnt6, which is expressed in the same cells as Wg throughout the development of the wing disc in both early and late stages, has no major role in wing fate specification but contributes to driving compensatory proliferation upon tissue damage and to producing tumour-like overgrowths upon CIN induction and additional blockage of the apoptotic pathway (Fig. 8). The mitogenic activity of Wnt6 was also identified in wing discs during normal development but only upon the depletion of Wg, the main driver of proliferative growth in these cells[18]. These data indicate that Wg activity levels might be saturated during normal development and that Wnt6 acts as a positive modulator of Wg only when a sharp increase in its activity levels is required to drive compensatory proliferation under stress conditions.

## Methods

### Fly maintenance, husbandry and transgene expression

Strains of *Drosophila melanogaster* were maintained on standard medium (4% glucose, 55 g/L yeast, 0.65% agar, 28 g/L wheat flour, 4 ml/L propionic acid and 1.1 g/L nipagin) at 25 °C in light/dark cycles of 12 h. The sex of experimental larvae was not considered relevant to

this study and was not determined. The strains used are summarised in Supplementary Table 8.

## Expression of reporter lines in development

Flies carrying the corresponding lacZ reporters were allowed to lay eggs for 24 h at 25 °C. For those experiments aimed at monitoring the developmental dynamics in expression pattern, flies were allowed to lay eggs for 6 h. Flies carrying GAL4/UAS transgenes were switched to 29 °C or were otherwise kept at 25 °C. Second instar (L2) or early third instar (eL3), mild third instar (mL3) and late third instar (L3) larvae were dissected at day 3, 4 or 5, respectively.

## Expression of reporter lines in regeneration

In the case of Eiger-induced cell death, females of the following genotype *wg¹-lacZ/CyO,GFP; rotund-GAL4,tubulin-GAL80ts,UAS-egr/ TM6B,tubulin-GAL80* were crossed with males of the following genotype: *GFP-Wg* or *nls-wnt6-GFP* and allowed to lay eggs for 24 h at 18 °C. Developing animals were kept at 18 °C until day 7 when they were switched to 29 °C for 16 h and dissected immediately after to isolate wing discs for immunostainings. In the case of Reaper-induced cell death, females of the following genotypes *wg¹-lacZ/ CyO,GFP; spalt-lexA,tub-Gal80ts*, or *GFP-Wg/CyO,GFP; spalt-lexA,tub-Gal80ts*, or *nls-wnt6-GFP/CyO,GFP; spalt-lexA,tub-Gal80ts* were crossed with males carrying the *lexO-reaper* transgene (genotype: *lexO-rpr/Cyo; MKRS/TM6B*) and allowed to lay eggs for 24 h at 18 °C. Developing animals were kept at 18 °C until day 7 when they were switched to 29 °C for 11 h and dissected immediately after to isolate wing discs for immunostainings.

## Regeneration experiments

Flies were allowed to lay eggs for 6 h at 18 °C. Developing animals of the following genotypes: (1) *+/+; rotund-GAL4,tubulin-GAL80ts,UAS-egr/+*, (2) *Δβ/Δβ; rotund-GAL4,tubulin-GAL80ts,UAS-egr/+* (3) *ΔƔ/ΔƔ; rotund-GAL4,tubulin-GAL80ts,UAS-egr/+*, (4) *ΔβƔ/ΔβƔ; rotund-GAL4,tubulin-GAL80ts,UAS-egr/+*; (5) *+/+; rotund-gal4, tubulin-GAL80ts, UAS-egr/UAS-egfp-i*, (6) *+/+; rotund-gal4, tubulin-GAL80ts, UAS-egr/UAS-wnt6-i*, (7) *Δβ/ΔƔ; rotund-GAL4, tubulin-GAL80, UAS-egr/+*, were kept at 18 °C until day 7 when they were switched to 29 °C to induce *eiger* expression for 16 h. Larvae were either dissected immediately after to isolate wing discs for immunostainings to analyse Wg intensity, number of pH3-positive cells and EdU incorporation, or returned to 18 °C until adulthood to allow regeneration. Experimental flies and control individuals were grown in parallel.

## Standard induction of CIN

Flies were allowed to lay eggs on standard fly food for 24 h (for the expression of reporter lines) or 8 h (for the quantification of tissue size) at 25 °C, larvae kept at 25 °C for an additional day, switched to 29 °C and dissected 4 days thereafter. Experimental flies and control individuals were grown in parallel. The larval genotypes used in Fig. 7 were the following: (1) *wg¹-lacZ/ wg:GFP; hh-GAL4 /UAS-rod-RNAi, UAS-p35* (Fig. 7a); (2) *wg¹-lacZ/ GFP-Wnt6; hh-GAL4/UAS-rod-RNAi, UAS-p35/+* (Fig. 7a); (3) *en-GAL4/reporter-lacZ; UAS-rod-RNAi, UAS-p35/+* (Fig. 7b); (4) *UAS-bsk-DN/+; en-GAL4/reporter-lacZ; UAS-rod-RNAi, UAS-p35/+* (Fig. 7c); (5) *+/+; hh-GAL4/UAS-rod-RNAi, UAS-p35* (Fig. 7d, e); (6) *Δβ/Δβ; hh-GAL4/UAS-rod-RNAi, UAS-p35* (Fig. 7d, e); (7) *ΔƔ/ΔƔ; hh-GAL4/UAS-rod-RNAi, UAS-p35* (Fig. 7d, e); (8) *ΔβƔ/ΔβƔ; hh-GAL4/UAS-rod-RNAi, UAS-p35* (Fig. 7d, e); and (9) *wnt6KO/wnt6KO; hh-GAL4/UAS-rod-RNAi, UAS-p35* (Fig. 7f, g).

## Immunohistochemistry

Wing, haltere, and leg imaginal discs, brains and ring glands of the indicated larval stage were dissected in cold PBS, fixed in 4% formaldehyde for 20 min, washed three times in PBT (PBS1%, 0.2% Triton), blocked for 45 min in BBT (PBS1X, 0.3% BSA, 0.2% Triton, 250 mM

NaCl), and incubated overnight with the following antibodies (see also Supplementary Table 8): mouse anti-MMP1 (1:50; 14A3D2, Developmental Studies Hybridoma Bank, DSHB); goat polyclonal anti-GFP (1:300; ab6673, Abcam); rabbit anti-β-galactosidase (1:600; 0855976, Cappel (MP Biochemicals)); mouse anti-β-galactosidase (40.1a, DSHB); rabbit anti-PH3 (1:1000; Merk Millipore); rat anti-Ci (1:10; 2A1, DSHB); mouse anti-Wg (1:50; 4D4, DSHB); mouse anti-Nubbin (1:50; nub2D4, DSHB); and rabbit anti-Tsh (1:100) kindly provided by S. Cohen. Discs were rinsed with BBT and incubated with secondary antibodies [Cy2, Cy3 and Cy5 (1:400), Jackson ImmunoResearch] and DAPI for 90 min. After 4 washes with PBT, discs were kept on mounting media (80 ml glycerol + 10 ml PBS 10x + 0.8 ml N-propyl-gallate 50%). The most representative images are shown in all experiments. At least 10–15 wing discs per genotype were imaged.

## DNA synthesis

Click-iT™ Plus EdU Alexa Fluor™ 647 Imaging Kit from Invitrogen (C10640) was used to measure DNA synthesis (S phase) in regenerating wing discs, following the manufacturer's indications. EdU (5-ethynyl-2′-deoxyuridine) provided in the kit is a nucleoside analogue of thymidine and is incorporated into DNA during active DNA synthesis. Time of incubation with EdU: 10 min. Experiments were carried out 3 consecutive days and 23–27 wing discs per genotype were imaged.

## Analysis of sequence conservation

Conservation of the CRMs spanning the *wg¹*-enhancer (*Alpha*, *Beta*, *Gamma* and *Delta*, Supplementary Fig. 2) was performed at the University of California Santa Cruz (UCSC) Genome Browser on *Drosophila melanogaster* (BDGP Release 6). In brief, multiple alignments of 27 insect species (23 *Drosophila* species, and *Musca Domestica*, *Anopheles gambiae*, *Apis melifera* and *Tribolium castaneum*) and measurements of evolutionary conservation used two methods (phastCons[41] and phyloP) from the PHAST package (http://compgen.cshl.edu/phast/), for all 27 species. Multiz and other tools in the UCSC/ Penn State Bioinformatics comparative genomics alignment pipeline (http://www.bx.psu.edu/miller_lab/) were used to generate multiple alignments. For more details, see description of methods at https:// genome-euro.ucsc.edu/index.html.

## Generation of *lacZ*-reporter lines

Different regions of the *wg¹*-containing enhancer named *Alpha* to *Delta*, *wnt6 Intron* and *SpdFlag* (Supplementary Table 1) were amplified by PCR from genomic DNA extracted from *w¹¹¹⁸* flies, using the suitable primers detailed in Supplementary Table 8. PCR products were digested with EcoRI/KpnI or KpnI/NotI restriction enzymes, gel-purified (NZYGelpure) and ligated into the pHs43nLacz vector previously digested with the same enzymes. Final constructs were checked by restriction, sequenced and sent for injection for transgenic generation in the *w¹¹¹⁸* background. At least six independent insertions per construct were analysed and shown to drive a reproducible expression pattern.

## Prediction of transcription factor binding sites

Predictions of binding sites in the *wg¹* region (see Supplementary Table 1 for coordinates) were performed using the Matscan software[42]. Position weight matrices were obtained from JASPAR 2020-all organisms[43], and MEME v12.21 Fly Factor Survey collection (https:// meme-suite.org/meme/db/motifs). Matrix logos were plotted using the seqLogo R package (https://bioconductor.org/packages/release/ bioc/html/seqLogo.html). Conservation score was taken from the evolutionary conservation track from UCSC (https://genome.ucsc. edu/index.html) for 26 species close to *Drosophila melanogaster* (DM6). Method of computation for conservation score: phastCons[41]. *p*-values were computed by permutation test in 1000 random genomic sequences of the same length as the analysed query sequence(s).

Predicted binding sites that overlapped using the same position weight matrix were merged and presented in Supplementary Tables 2–5, where maximum and average match score (score.max, score.avg), maximum and average conservation score (cons.max, cons.avg), and minimum and maximum permutation test $p$-value (pv.min, pv.max) for all individual hits within the overlapping region are shown.

### Generation of *Gamma* and *Gamma*-590 reporters carrying mutations in the Ci binding site

To introduce the Ci mutation in the *Gamma*-LacZ reporter, two independent PCRs were performed using the pHsnLacZ-*Gamma* as a template. The first PCR introduced the mutation GCGT**GTG**GTCT → GCGT**ATA**GTCT in the Reverse primer (see PCR1 wg-*Gamma*Ci mut on Supplementary Table 8) and the second PCR introduced the same mutation on the Fwd primer (see PCR2 wg-*Gamma*Ci mut on Supplementary Table 8). A third PCR was performed on the mixed PCR1 and PCR2 products using the wg-*Gamma* Fwd and wg-*Gamma* Rev primers. The final product was 1145 bp long bearing the mutation in the Ci binding site. The wg-*Gamma*-mut was digested and cloned into the pHsnLacZ vector. To obtain the *Gamma*-590Cimut-LacZ reporter, a PCR with the *Gamma*590-Fwd and *Gamma*-Rev was performed using the pHsnLacZ-*Gamma*Cimut construct as a template. The PCR product was digested and cloned in the pHsnLacZ vector. Final constructs were checked by restriction, sequenced and sent for injection for transgenic generation in the $w^{1118}$ background.

### Generation of *wg¹*-reporter carrying mutations in AP1 binding sites

Five AP1 binding sites were present in the *wg¹*-enhancer and AP1 binding sites 1 to 4 (oriented 5' to 3' in Fig. 6a) were mutated (labelled with stars in Fig. 5a). These four AP1 binding sites were mutated introducing XbaI restriction sites in oligos. For sites 1 and 4 the following mutations were introduced: CGCGCTTATGTTTCTA**TGATTC**AGCAGCCAGATT → CGCGCTTATGTTTCTA**TCTAGA**AGCAGCCAGATT. TCTCTGCTGGCTGACGT**TTAGTC**ATAAAATATTCCA → TCTCTGCTGGCTGACGT**TCTAGA**ATAAAATATTCCA. The two oligos were annealed to vector pBS-wg¹ followed by PCR polymerase reaction, Klenow blunt, ligation, DpnI digestion and transformation. Minipreps were checked by XbaI restriction. Mutations 2 and 3 were introduced performing independent PCRs using pBSK-wg¹ Mut1&4 as a template. For Mutation 2 the first PCR introduced the mutation TAGC**TGACTC**ACTC → TAGC**TCTAGA**ACTC on the Reverse primer and the second PCR introduced the same mutation on the Forward primer. A third PCR was performed on the mixed PCRs products using wg-AP1 Fwd and wg-AP1 Rev primers. For Mutation 3 the same strategy was used introducing mutation AGTC**TGACTA**ATAC → AGTC**TCTAGA**ATAC. The final vector pBSK-wg¹ Mut1,2,3&4 was sequenced, digested KpnI/NotI and cloned into the pHsnLacZ vector.

### Generation of *wg¹*-enhancer deletions with the CRISPR/Cas9 technique

gRNAs Up and Down to generate the different *wg¹*-enhancer deletions were cloned in the pBFv-U6.2B vector in three steps as follows:

1. gRNA_Up was designed, and sticky ends for BbsI were added to the 5' end of the Fwd (CTTC) and Rev (AAAC) oligos (see Supplementary Table 8). 9.5 μl of Fwd and Rev oligos at a concentration of 100 μM were mixed with 1 μl of SSC20X, boiled for 5 min and allowed to cool down overnight in a 1 L water bath for efficient annealing. 1 μl of 1/20 dilution of the mix was used to ligate with the pBFv-U6.2 plasmid previously digested with BbsI. Ligation was transformed into DH5Alpha competent bacteria and 3 colonies were selected for DNA miniprep and sequencing with the T3 primer.

2. gRNA-Down was designed, and sticky ends for BbsI were added to the 5' end of the Fwd (CTTC) and Rev (AAAC) oligos (see

Supplementary Table 8). Fwd and Rev oligos were annealed and ligated in the pBFv-U6.2B plasmid previously digested with BbsI. Ligation was transformed and 3 positive colonies were selected for checking and sequencing.

3. pBFv-U6.2-gRNA_Up was digested with EcoRI/NotI and the gRNA_Up insert was gel-purified and ligated with the pBFv-U6.2B-gRNA_Down, previously digested with EcoRI/NotI. Five colonies were selected for plasmid preparation and sequenced with the T3 primer.

One positive colony pBFv-U6.2B-gRNA_Up+Down was selected for Maxiprep and the DNA was used for injection into $y^1v^1P\{nos\text{-}phi\text{-}C31\backslash int.NLS\}X$; $P\{CaryP\}attP40$ (BL 25709) flies. Trangenics were identified as $v+$ individuals.

gRNAup+down transgenic flies were then crossed to $y^1$ *cho* $v^1$; attP40.nosCas9/Cyo flies to generate the deletions in the germline. Five males from the progeny that carried gRNAUp+Down and the Cas9 were individually crossed with $v^1$; Sco/SM6a (BL137) females. In a second step, 5 to 10 males from each cross were selected and individually backcrossed with $v^1$; Sco/SM6a females. 2–3 individuals from each established stock were used for genomic extraction and PCR checking to identify the mutant lines (see Supplementary Table 8).

### Molecular characterisation of deletions

To identify CRISPR mutants, genomic DNA was extracted from individuals of 10–15 different candidate lines. For this purpose, 2–3 flies from each line were squashed in 300 μL of homogenizing buffer (0.1 M Tris-HCl pH 9, 0.1 M EDTA, 1% SDS) and the mix incubated at 65 °C for 30 min. After incubation, 67.8 μl of KAc was added and tubes were kept on ice for 30 min. Samples were centrifuged for 10 min at 4 °C, the supernatant was transferred to a new tube and DNA was precipitated by adding 0.5 volumes of isopropanol, incubated 5 min at RT and centrifuged again for 5 min at RT. The pellet was washed with 70% EtOH and resuspended in 50 μl of TE. 1 μl was then used to perform a PCR with the suitable primers (see Supplementary Table 8). PCR products from samples that showed the expected deletions were gel-purified and sent for Sanger sequencing to characterise indels.

### Quantification

(1) Wing to notum transformation: each heminotum was considered an independent event. The percentage of wing to notum transformation (loss of wings and appearance of an ectopic heminotum where bristles showed reverse polarity) was calculated by dividing the number of transformed heminota by the total number of heminota (% wing to notum transformation = ($n$ transformed heminota/$n$ total heminota) × 100). Representative pictures of the phenotype were obtained.

(2) Adult wing regeneration: Adult flies were collected in SH buffer (75% glycerol, 25% ethanol) and wings were dissected in water and mounted in Faure's mounting medium. Regenerated wings were scored using Fiji Software (NIH, USA). They were considered regenerated when they were capable of reaching a regular size. No patterning problems were considered when assessing the regenerative potential. The percentage of regenerated wings was calculated by dividing the number of regenerated wings by the total number of scored wings (% regenerated wings = ($n$ regenerated wings/$n$ total wings) × 100).

(3) Wing disc size upon CIN: The sizes of the whole wing disc (based on DAPI staining) were measured manually using Fiji Software (NIH, USA) on Z-projection of the wing disc obtained using a Zeiss LSM780 confocal microscope at ×25 glycerol immersion objective with 1.5 μm per optical section covering the entire thickness of each disc.

(4) Wingless signal intensity: Wingless areas in the pouch of regenerating wing discs and in the pouch and hinge of the P compartment of wing discs subjected to CIN (based on the absence of Ci expression) were selected using the polygonal tool of Fiji Software (NIH, USA). Wingless intensity (in arbitrary units, a.u.) was measured upon setting a fluorescence threshold for the corresponding channel.

Image stacks were obtained using a Zeiss LSM780 confocal microscope at ×40 oil immersion objective with 1.5 μm per optical section for regenerating discs, and at ×25 glycerol immersion objective with 1.5 μm per optical section for wing discs subjected to CIN. The entire thickness of each disc was covered in both cases. Maximum intensity Z-projection was performed on the stacks prior to quantification.

(5) Mitotic activity: Mitosis was measured by counting mitotic cells (pH3-positive cells) present in an area slightly broader to the one of the Wg expression domain (red lines in Fig. 6c) of regenerating wing discs using the Fiji Software (NIH, USA). Image stacks were obtained using a Zeiss LSM780 confocal microscope under a ×40 oil immersion objective with 1.5 μm per optical section to cover the entire thickness of each disc. The ratio between the number of mitotic cells and Wg area (sizes measured in arbirtrary units, a.u., using the polygonal tool on Fiji) was calculated.

(6) EdU incorporation: The region comprising the wing pouch and hinge primordia was selected using the polygonal tool of Fiji and the area was quantified. EdU positive area within this region was measured using a Macro created in Fiji. The ratio between the areas of EdU incorporation and wing pouch and hinge regions (sizes measured in arbirtrary units, a.u.) was calculated. Experiments were carried out in parallel in all the genotypes analysed and experiments were repeated three times.

## Microscopy
Larval discs or tissues were analysed and scanned with a LSM 780 Zeiss confocal microscope. Adult wings, nota and eyes were analysed and pictured with an Olympus MVX10 macroscope. Regenerating wings were imaged using an ECLIPSE E600 microscope coupled to a NIKON DSRi2 camera.

## Statistics and reproducibility
The statistical analysis for comparison of means was performed by linear regression using the experimental batch as adjusting variable. Normality assumption was tested for every fitted model, applying a log2-transformation of the data when necessary. However, for clarity of representation, data are shown in the original scale. The statistical analysis for comparison of percentages of regenerated wings was performed by logistic regression using the experimental batch as adjusting variable. In both types of models, Dunnett's multiple comparisons correction was applied when comparing the means/percentages of several experimental groups with a common control. Differences were considered significant when adjusted $p$ values were <0.001 (***), 0.01 (**) or 0.05 (*). Statistical analysis was carried out with the multcomp[44] R package. All genotypes included in each histogram or scatter plot were subjected to the same experimental conditions (temperature and time of transgene induction) and analysed in parallel, and all experimental quantifications were carried out at least three times in different days with at least 12 wing discs, 27 adult wings and 22 heminota per genotype. Information about the $n$ values, $p$ values, and statistical tests used can be found in the figure legends and in Supplementary Tables 6, 7. Representative micrographs of at least 10 different wing discs of the indicated age, genotype and immunostaining are shown in the figures.

## Reporting summary
Further information on research design is available in the Nature Research Reporting Summary linked to this article.

## Data availability
Source data are provided with this paper.

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

## Acknowledgements

We thank I. Guerrero, I. Hariharan, R.E. Harris, J.P. Vincent, G. Schubiger, F. Serras, A. Teleman, and the Bloomington *Drosophila* Stock Center (USA), the Vienna Drosophila Resource Center (Austria), and the Developmental Studies Hybridoma Bank (USA) for flies and antibodies, Lara Barrio and Elena Fusari for comments on the manuscript, and the IRB Advanced Digital Microscopy (ADM) and Biostatistics and Bioinformatics Facilities for support. This work was funded by the *BFU2016-77587-P* and *PID2019-110082GB-I00* grants to M.Milán from the Spanish Ministry of Science and Competitiveness, and ERDF "Una manera de hacer Europa". We gratefully acknowledge institutional funding from the Spanish Ministry of Science and Competitiveness through the Centres of Excellence Severo Ochoa Award, and from the CERCA Programme of the Government of Catalonia.

## Author contributions

E.G., L.P., M.Muzzopappa and M.Milán conceived and designed the experiments, E.G., L.P. and M.Muzzopappa performed the experiments, all authors analysed the data, and M.Milán wrote the paper.

## Competing interests

The authors declare no competing interests.
