## [Peer Review File · Nature Communications]

A single WNT enhancer drives specification and regeneration of the *Drosophila* wingEditorial Note: Parts of this Peer Review File have been redacted as indicated to remove third-party material where no permission to publish could be obtained.

REVIEWER COMMENTS

Reviewer #1 (Remarks to the Author):

In this study, Gracia-Latorre et al. dissect the regulatory mechanisms that control the expression pattern of *wingless*, a pleiotropic and *Drosophila*-favorite gene, both during the normal development of several organs, and also in contexts causing a proliferative boost, like regeneration and tumorous growth. Combining bioinformatic analysis of putative binding sites, reporter lines and mutants, they sub-divide the *wg1* enhancer in fragments and then identify two cis-regulatory modules (CRMs) controlling its expression on wings. These CRMs act redundantly, but show relevant differences in the signals capable of activating them. The authors finally find that these two CRMs are also capable of responding to the stress signal JNK during wing regeneration after genetic ablations, and during the tumorous growth caused by Chromosomal Instability.

The manuscript addresses an interesting biological topic and serves as a model of how genes are differentially regulated at the enhancer level during normal contexts versus supervened conditions. It is well defined on its scope and well executed. Data is correctly presented, controls are wisely chosen and the interpretation of results and conclusions fit in general the presented data. Notably, the study has generated a collection of reagents that will be of great interest in the fly community for the study of processes where Wnt-family genes are implicated. Overall, I consider the manuscript has high value, and while I have some comments, these issues are minor on its nature and can be corrected mostly with text/figure modifications, with the exception of the first two points that will require additional (but straight-forward) experiments.

- The activation of the Beta and Gamma CRMs during regeneration (Fig 6B), as well as their requirement for regenerating the wing blade (Fig 6G) are nicely and abundantly documented by the authors. The explanation for the lack of regeneration on Beta and Gamma mutants being caused by a reduction on proliferation (as opposed to other possibilities as a role in correct wing re-patterning) is not as solid experimentally, in the other hand. The authors compare quantifications of a mitosis marker, pH3, to conclude that proliferation is reduced. While this is a valuable result pointing to their conclusion, the problem with pH3 is that it is a mere "snapshot" of the cell cycle that can't discern if cells are cycling faster, or rather cells are spending more time in M-phase but overall are proliferating at the same rate. To address this, the authors should confirm their results with either a clonal analysis or an S-phase labelling. As S-phase labeling protocols can be technically challenging, I suggest the authors using the "Invitrogen Click-iT EdU Cell Proliferation Kit for Imaging", that we have used routinely with success for several different *Drosophila* tissues.
- The results about the role of *wnt6* during regeneration (affecting regeneration capacity and proliferation) and tumoral growth are novel, relevant and well supported. The conclusions about its role modulating *wg* expression, on the other hand, are not sufficiently supported by the experiments. The effects during regeneration (Fig 6I) are assessed using an RNAi, which could have *wg* as an off-target, explaining the reduced *wg* levels as a nonspecific effect. Ideally, the experiment should be confirmed using the *wnt6*KO allele used for the tumoral growth experiments (Fig 7G). Both experiments present, however, a subsequent problem: the *Wg* antibody used to quantify expression could be recognizing the Wnt6 protein nonspecifically. An alternative approach could be measuring *wg* transcript levels by qPCR of isolated discs. In any case, the conclusion of Wnt6 modulating *Wg* levels is not a core part of the manuscript, and its quality won't be reduced if the authors decide to leave the results out for future investigation.

Other minor points:

- Figure 3C presents an additional reporter line, *gamma-delta-lacZ*, that is not mentioned anywhere else on the text. The reasoning and conclusions about it should be explained, or otherwise the panels removed.
- Line 163 indicates that that the mutation of a Ci binding site compromised "the ability of Gamma

and the 590 bp-long fragment to drive sustained and restricted expression of lacZ to the ventral anterior wedge of early wing discs". Fig 3D shows, however that the phenotype is instead manifested in late wing discs. Can the authors clarify what they mean here?

- Fig 3D shows that the gamma-630 does not drive wg expression, while the gamma-590 reporter recapitulates the expression pattern of full gamma. Meanwhile, Fig 4C shows a phenotype considerably stronger for Δ gamma trans-heterozygotes than for Δ gamma-590 trans-hets. Can authors elaborate on possible explanations?
- Line 197 states that beta only responds to the negative input of Vn, and not of Hh signaling. The fact that beta-lacZ is not compartment-restricted supports the hypothesis, but a functional experiment on shown on Fig 3F, such as $sd>Ptc$ (similar to what is shown for gamma on Fig 3G), would be desirable.
- Line 224 indicates that alpha, beta and gamma are expressed in leg discs. However, Figure S3 does not show activity of beta on neither the leg nor the antenna part of the disc.
- Line 277 states that mitotic activity was significantly reduced in beta, gamma and beta-gamma mutants. But Fig 6D shows "NS" for beta. This doesn't change the global conclusion of the experiment, as gamma and beta+gamma are clearly reduced, but this should be clarified on the text.
- Fig 6B, alpha-LacZ panel depicts a wing disc with a normal outlook despite suffering ablation in the wing pouch by $rn>egr$. Do the authors have a more representative image or is this a normal outcome?
- The y axis of the charts in Fig 6D and 6H should show explicitly the units they measure. In the case of "signal intensity", are these absolute numbers yielded by the quantification software or are they relativized to a control? Is pH3 expressed as "number of pH3 cells per μm^2 " or something else?
- Figure 6D shows that PH3 is diminished in the beta and gamma mutants in the ablated region. As Wg has non-autonomous proliferative effects, the authors might want to consider analyzing the proliferation on a ring surrounding the ablated region as well.
- The quantifications of "% of regeneration" shown on Fig 6G could be, in some cases, confused with the normal wing-to-notum transformations caused by some of the allelic combinations. Are the authors certain the non-regenerated wings are caused by a specific role of these enhancers during regeneration, rather than the effect of adding up an ablation on top of a wing development effect? Would it be possible to present control bars without $rn>egr, G80ts$ in these charts?
- The authors found that the gamma-630 sub-fragment does not get activated during normal development (Fig 3D), but curiously gets activated during regeneration and tumoral growth (Figs 6 and 7). This is an interesting finding that should be highlighted on the discussion.

Salvador Herrera, PhD

Reviewer #2 (Remarks to the Author):

Gracia-Latorre et. al. investigated and characterized a specific WNT (wingless, wg) enhancer (i.e., wg1), and in particular two cis-regulatory modules (CRM) in *Drosophila melanogaster*. The authors combined classical genetics, confocal microscopy, CRISPR/Cas9 (for in-vivo deletions of candidate CRM) and regeneration & tumor models to show that wg1 allele's phenotype is caused by loss of a core 1.8 kb enhancer, which is functionally restricted to the developing wing primordium. Using CRISPR/Cas9 mediated deletions and lacZ reporter assays, the authors identified two conserved CRMs which function redundantly in a spatio-temporal manner and showed that these CRMs are also essential in the context of regeneration of the *Drosophila* wing.

While I think that the authors present interesting and important insights into wg1 enhancer-based regulation of the wg gene during development, regeneration and malignant growth, the paper (which is in most cases technically sound) currently lacks some clarity and fails at specific points to support their conclusions with the experiments performed. Major revisions are required before this comprehensive work can be published in Nature Communications. I understand that a lot of work has been done and thus only suggest additional experiments where I think that the the conclusions do not match the data presented.

Main comments

1) In lines 151-153 the authors state: "On the basis of sequence conservation with other *Drosophila* species, we subdivided a 5 kb-long region containing the wg1 deletion into four fragments [Alpha, Beta, Gamma, and Delta; Figure 3a] and generated reporter constructs carrying them." Throughout the manuscript the authors highlight and emphasize this conservation, however, they never show a sequence alignment or even report evidence that the characterized binding sites are part of these conserved elements. Moreover, the conservation has been the basis to subdivide the elements into the four fragments, they analysed and used in their experiments, two of which are stated to be the conserved regulatory modules (CRMs) and in fact the central message of the paper.

The details how this comparison amongst *Drosophila* species was performed should be described in the Materials and Methods section. In addition, the necessary sequence alignments (across species) as well as the conserved arrangement of the Ci, Ets, and AP-1 binding sites should be shown. Since the authors also perform experiments to show the relevance of the predicted binding sites and in part correlate their prediction score with their biological significance (high scoring being more relevant), also this aspect of conservation amongst different *Drosophila* species should be documented in the paper. In this respect, a Figure including the detailed information of all the fragments highlighting the position of the CRMs as well as the relative positioning of the individual TF binding sites (AP-1, Ci, ETS) within the CRM should be shown and labeled with an appropriate scalebar. Currently all this information is scattered amongst different Figures and Tables. Such a Figure would help the reader to get a better view on, and understanding of, the data. Also, a "model" Figure summarizing all the data, findings and conclusions, i.e. signals/cues, the involvement of the enhancer element wg1 (particularly the role of the two independent CRMs) and the respective outputs Wg/Wnt6 with respect to development, regeneration and cancer would be very helpful to draw a take home message and to put the experiments into a bigger context. Especially the discussion would profit from such a figure. Without this information, the subdivision of the wg1 enhancer into two CRMs and two less relevant units (alpha delta) as well as the emphasis on conservation of CRM function, is difficult to grasp and remains vague.

2) In the Introduction (lines 26/27) the authors mention, that: "We identify two evolutionary conserved cis-regulatory modules within this enhancer that are utilized in a redundant manner to mediate these two activities through the use of distinct molecular mechanisms". While I agree that the mechanisms/signaling events that lead to enhancer activation are shown to be different, some of the experiments do not entirely allow to draw this conclusion. In the context of regeneration and malignant overgrowth, the authors delete one or both of the CRMs to study their function in this context. Their claim is that the responses monitored are mediated by the AP-1 binding sites within the two CRMs. However, the monitored responses are reductions and not complete eliminations of Eiger-induced wg expression (lines 264-266; "The levels of Wg expression induced by Eiger were significantly reduced in Δ BetaGamma homozygous flies and partially reduced in flies where either the Gamma or Beta CRMs were deleted (Figure 6c, d)"), and thus it may be that slightly lower scoring AP-1 binding sites (as depicted in Table 2) remain in the alpha fragment (i.e., 3rd CRM?). In addition, the authors also delete all binding sites for ETS and Ci and possible other, so far unknown regulatory elements by deleting the entire CRMs. In fact, the authors mention (line 56ff) that "...in older larvae and once wing specification is underway, this enhancer remains silent but accessible to transcription factor binding in growing wing primordia". Therefore the authors cannot completely rule out that the observed effects are exclusively due to the elimination of AP-1 sites. To really prove that the AP-1 binding sites themselves are necessary and sufficient for the effects they observe, the authors should perform the following experiments: They should repeat a subset of the regeneration and malignant overgrowth experiments making the following changes. Rather than assessing Wg expression in the Δ BetaGamma deletion setting, the authors should use wg1-lacZ reporter where exclusively (some of) the AP-1 sites are mutated, in either or both of the CRMs leaving the rest of the CRMs intact. The

responses can be compared to the wild-type *wg1-lacZ* transgenes already available. Also, the authors might consider making AP1 deletions (with and without *Ci* deletions) in the context of *wg1-lacZ* reporter transgene – that way the authors could better distinguish between the role of these CRMs (and their redundancies) in development (Hh signaling, early expression and maintenance of *Wg*) vs regeneration / stress (*CIN* cancer model).

Minor comments

In lines 125-127, the authors state that "Surprisingly, we observed that expression of *wg1-lacZ* was not abolished upon *Ptc* overexpression (Figure 2g, arrows), suggesting that Hh signalling is not an absolute requirement for the expression of *Wg* in early wing discs."

While this is a very likely scenario, an alternative explanation may be possible: It could be, that the *ptc* overexpression via *sd-Gal4* cannot completely abolish *wg1-lacZ* expression because the scalloped expression domain may not entirely cover all the cells of the *wg-lacZ* expression domain in the second instar larvae. Unfortunately I did not find this data in the manuscript. Even after a literature search it remains unclear how *sd* is expressed in a 1st or 2nd instar wing disc. Importantly, this should be shown in the paper as the function of the *wg1* enhancer is studied in this context and not the 3rd instar larvae. Given the dynamic changes of *wg* expression in 1st, 2nd and 3rd instar larvae *sd-Gal4* does not necessarily completely overlap with the *wg*-expression domain. Confusing is also the labelling of Figure 2g: In the Figure legend it is mentioned that the larvae expressed the indicated transgenes under the control of the *sd-gal4* driver. However, the panel is entitled *sd>EGFP* the green staining should however show *Ci* expression. Please clarify the labelling of the Figure.

Similarly to the *sd-Gal4* expression domain, the authors should shortly explain in the text why they are using *rd-Gal4* constructs and how they are expressed, as the paper should be easily accessible also to a more broader audience that does not have a *Drosophila* background. The reader should not be forced to read all the previously published papers to figure out that this line has been used and described elsewhere (i.e., Ref 13).

Finally, the authors may consider changing the current title " Sequential activities of a WNT enhancer in specification and regeneration of the *Drosophila* wing". Sequential implies that the events follow one another which must not necessarily be the case. On the other hand, both events (specification and regeneration) may accidentally happen at the same time - which may provoke the mentioned discrepancies in the findings (mentioned in line 237-239). For this reason, the authors refine their experimental settings (lines 252-254). I believe that the current title is not well chosen.

REVIEWER COMMENTS

Reviewer #1 (Remarks to the Author):

In this study, Gracia-Latorre et al. dissect the regulatory mechanisms that control the expression pattern of wingless, a pleiotropic and drosophilists-favorite gene, both during the normal development of several organs, and also in contexts causing a proliferative boost, like regeneration and tumorous growth. Combining bioinformatic analysis of putative binding sites, reporter lines and mutants, they sub-divide the wg1 enhancer in fragments and then identify two cis-regulatory modules (CRMs) controlling its expression on wings. These CRMs act redundantly, but show relevant differences in the signals capable of activating them. The authors finally find that these two CRMs are also capable of responding to the stress signal JNK during wing regeneration after genetic ablations, and during the tumorous growth caused by Chromosomal Instability.

The manuscript addresses an interesting biological topic and serves as a model of how genes are differentially regulated at the enhancer level during normal contexts versus supervened conditions. It is well defined on its scope and well executed. Data is correctly presented, controls are wisely chosen and the interpretation of results and conclusions fit in general the presented data. Notably, the study has generated a collection of reagents that will be of great interest in the fly community for the study of processes where Wnt-family genes are implicated. Overall, I consider the manuscript has high value, and while I have some comments, these issues are minor on its nature and can be corrected mostly with text/figure modifications, with the exception of the first two points that will require additional (but straight-forward) experiments.

We appreciate reviewer's comments on our work. We have been able to address all reviewer's concerns with additional experiments and changes in the text and in the figures (see below).

- The activation of the Beta and Gamma CRMs during regeneration (Fig 6B), as well as their requirement for regenerating the wing blade (Fig 6G) are nicely and abundantly documented by the authors. The explanation for the lack of regeneration on Beta and Gamma mutants being caused by a reduction on proliferation (as opposed to other possibilities as a role in correct wing re-patterning) is not as solid experimentally, in the other hand. The authors compare quantifications of a mitosis marker, pH3, to conclude that proliferation is reduced. While this is a valuable result pointing to their conclusion, the problem with pH3 is that it is a mere "snapshot" of the cell cycle that can't discern if cells are cycling faster, or rather cells are spending more time in M-phase but overall are proliferating at the same rate. To address this, the authors should confirm their results with either a clonal analysis or an S-phase labelling. As S-phase labeling protocols can be technically challenging, I suggest the authors using the "Invitrogen Click-iT EdU Cell Proliferation Kit for Imaging", that we have used routinely with success for several different Drosophila tissues.*

As suggested by this reviewer, we have included EdU incorporation experiments in Figure 6c and 6d to complement our pH3 quantification experiments and conclude, as stated in pg12 of the ms that (underlined) "Consistently, Wg expression levels and mitotic activity in wing discs subjected to Eiger expression, which is increased when compared to controls⁴, were reduced in $\Delta\gamma$, $\Delta\beta$ and $\Delta\beta\gamma$ homozygous individuals (Figure 6c, d). Similar results were obtained when monitoring cells in S-phase (Figure 6c, d)."

We really appreciate reviewer's suggestion.

- The results about the role of wnt6 during regeneration (affecting regeneration capacity and proliferation) and tumoral growth are novel, relevant and well supported. The conclusions about its role modulating wg expression, on the other hand, are not*

sufficiently supported by the experiments. The effects during regeneration (Fig 6I) are assessed using an RNAi, which could have *wg* as an off-target, explaining the reduced *wg* levels as a nonspecific effect. Ideally, the experiment should be confirmed using the *wnt6*KO allele used for the tumoral growth experiments (Fig 7G). Both experiments present, however, a subsequent problem: the *Wg* antibody used to quantify expression could be recognizing the *Wnt6* protein nonspecifically. An alternative approach could be measuring *wg* transcript levels by qPCR of isolated discs. In any case, the conclusion of *Wnt6* modulating *Wg* levels is not a core part of the manuscript, and its quality won't be reduced if the authors decide to leave the results out for future investigation.

We agree with this reviewer that the impact of *Wnt6* depletion on *Wg* protein levels is not a core part of the manuscript. Based on the fact that many more experiments will be required to support our proposal, we have decided, as suggested by this reviewer, to leave the results out for future investigation. We would like to point out that the *Wg* antibody does not detect *Wnt6* protein in overexpression experiments (See figure below) and that the *wnt6*-RNAi does not appear to off target *wg* because this line was shown not to cause any overt phenotype by its own when targeted to the developing wing (see Barrio & Milán, 2020).

[redacted]

Other minor points:

- Figure 3C presents an additional reporter line, *gamma-delta-lacZ*, that is not mentioned anywhere else on the text. The reasoning and conclusions about it should be explained, or otherwise the panels removed.

In page 8 (now labeled in red), we mention the use of *gamma-delta-lacZ* as follows: “We noticed that expression of *lacZ* driven by *Beta* and *Gamma* CRMs in second instar wing discs persisted in later stages even in the presence of *Delta* (Figure 3c), a genomic region that has been shown to block the activity of this enhancer upon tissue damage in mature wing discs (13). These results suggest that the regulatory elements responsible for turning *Gamma* and *Beta* CRMs off during normal development are not present in these genomic regions.”

- Line 163 indicates that that the mutation of a *Ci* binding site compromised “the ability of *Gamma* and the 590 bp-long fragment to drive sustained and restricted expression of *lacZ* to the ventral anterior wedge of early wing discs”. Fig 3D shows, however that the phenotype is instead manifested in late wing discs. Can the authors clarify what they mean here?

We appreciate reviewer's comment. We have changed the images of panel Figure 3d with new examples of second instar wing discs where the expression of *lacZ* in the posterior compartment (marked by green arrowheads) and the broader expression of *lacZ* in the anterior compartment are more visible.

- Fig 3D shows that the gamma-630 does not drive wg expression, while the gamma-590 reporter recapitulates the expression pattern of full gamma. Meanwhile, Fig 4C shows a phenotype considerably stronger for Δ gamma trans-heterozygotes than for Δ gamma-590 trans-hets. Can authors elaborate on possible explanations?

As suggested, we have added the following sentences in pg 9 of the revised ms: “We observed that the penetrance of the wingless adult phenotype was higher in $\Delta\gamma$ than in Δ 590 transheterozygous (Figure 4c) pointing to a potential role of the remaining 630 bp-long fragment in the regulation of wingless expression despite its inability to drive lacZ expression (Figure 3d). Whether this fragment contributes to transcription factor accessibility, enhancer-promoter interactions or chromatin conformation remains to be further investigated. “

- Line 197 states that beta only responds to the negative input of Vn, and not of Hh signaling. The fact that beta-lacZ is not compartment-restricted supports the hypothesis, but a functional experiment on shown on Fig 3F, such as $sd>Ptc$ (similar to what is shown for gamma on Fig 3G), would be desirable.

We have included new experimental data to address this issue in Figure 3f. As now indicated in the ms (pg 8, underlined): “Despite the presence of a predicted low-score Ci binding site in Beta (light green bar in Figure 3a, b and Table S1), this fragment drove expression of lacZ to both A and P cells in wild type larvae (Figure 3c), and this expression was not expanded throughout the A compartment upon ubiquitous expression of Hh (Figure 3f), thereby indicating that this binding site is not functionally relevant in this context.”

- Line 224 indicates that alpha, beta and gamma are expressed in leg discs. However, Figure S3 does not show activity of beta on neither the leg nor the antenna part of the disc.

We apologize for this mistake and have removed Beta from the sentence as follows (please note that Figure S3 is now S4): “Although Alpha and Gamma were expressed in the leg discs (Figure S4d, f), no overt phenotype was observed in the adult appendage of wg1 mutant flies (not shown).”

- Line 277 states that mitotic activity was significantly reduced in beta, gamma and beta-gamma mutants. But Fig 6D shows “NS” for beta. This doesn’t change the global conclusion of the experiment, as gamma and beta+gamma are clearly reduced, but this should be clarified on the text.

We agree and have removed the word “significantly” from the sentence: “Consistently, Wg expression levels and mitotic activity in wing discs subjected to Eiger expression, which is increased when compared to controls⁴, were reduced in $\Delta\gamma$, $\Delta\beta$ and $\Delta\beta\gamma$ homozygous individuals (Figure 6d).”

- Fig 6B, alpha-LacZ panel depicts a wing disc with a normal outlook despite suffering ablation in the wing pouch by $rn>egr$. Do the authors have a more representative image or is this a normal outcome?

We have included as Figure 6b a more representative image of an ablated wing disc, as suggested.

- The y axis of the charts in Fig 6D and 6H should show explicitly the units they measure. In the case of “signal intensity”, are these absolute numbers yielded by the quantification software or are they relativized to a control?

Wingless signal intensity values are absolute values and measured in arbitrary units (a.u.) as described in the Quantification sub-section of the Methods section. We have included these units in the corresponding figure panels and have added the units of quantification in the Methods section too.

Is pH3 expressed as “number of pH3 cells per μm^2 ” or something else?

The number of pH3 cells is expressed as the number of pH3 cells per area, this one measured in arbitrary units (a.u.). We have included these units in the corresponding figure panels and have added the units of quantification in the Methods section too.

• *Figure 6D shows that PH3 is diminished in the beta and gamma mutants in the ablated region. As Wg has non-autonomous proliferative effects, the authors might want to consider analyzing the proliferation on a ring surrounding the ablated region as well.*

We thank reviewer for this suggestion. Indeed, we quantified mitotic activity in the area limited by a red line in Figure 6c, which included those cells expressing Wg and a ring surrounding the ablated region. We have carried out the following changes to clarify this issue. First, we have included a reference to this panel in the figure legend as follows: “the area where mitotic activity was quantified in d, h is labeled by a red line in c.” Second, we have changed the y-labels in Figures 6d, i to avoid confusion as follows: “number of pH3 cells/area (a.u.)” instead of “number of pH3 cells/Wg area (a.u.)”. Third, we have included in the Methods section the following sentence: “Mitosis was measured by counting mitotic cells (pH3-positive cells) present in the area a couple of cell diameters broader than the Wg expression domain (red lines in Figures 6c) of regenerating wing discs using the Fiji Software (NIH, USA).”

• *The quantifications of “% of regeneration” shown on Fig 6G could be, in some cases, confused with the normal wing-to-notum transformations caused by some of the allelic combinations. Are the authors certain the non-regenerated wings are caused by a specific role of these enhancers during regeneration, rather than the effect of adding up an ablation on top of a wing development effect?*

We have included the following sentence in pag 11 to clarify this point: “In this case, those flies presenting a notum duplication and absence of wing tissue, as a result of the developmental role of both CRMs in wing fate specification, were not scored.”

Would it be possible to present control bars without $rn>egr, G80ts$ in these charts?

Based on the fact that those flies presenting a notum duplication and absence of wing tissue, as a result of the developmental role of both CRMs in wing fate specification, were not scored, we think this is not absolutely necessary and might confuse the reader.

• *The authors found that the gamma-630 sub-fragment does not get activated during normal development (Fig 3D), but curiously gets activated during regeneration and tumoral growth (Figs 6 and 7). This is an interesting finding that should be highlighted on the discussion.*

As pointed by this reviewer in a previous point, gamma-630 sub-fragment does not drive lacZ expression during normal development but it has a role in wing fate specification as the penetrance of the wingless adult phenotype is higher in $\Delta\gamma$ than in $\Delta 590$ transheterozygous (Figure 4c). These results point to a potential role of the 630 bp-long fragment also in the regulation of wingless expression despite its inability to drive lacZ expression (Figure 3d). The fact that the gamma-630 sub-fragment drove lacZ expression in regenerating discs but not during normal development is, most probably, a consequence of the presence of AP1 binding sites but not Ci binding sites. Thus, we do not think this issue should be highlighted in the Discussion. We appreciate this suggestion, though.

Salvador Herrera, PhD

Reviewer #2 (Remarks to the Author):

Gracia-Latorre et. al. investigated and characterized a specific WNT (wingless, wg) enhancer (i.e., wg1), and in particular two cis-regulatory modules (CRM) in Drosophila melanogaster. The authors combined classical genetics, confocal microscopy, CRISPR/Cas9 (for in-vivo deletions of candidate CRM) and regeneration & tumor models to show that wg1 allele's phenotype is caused by loss of a core 1.8 kb enhancer, which is functionally restricted to the developing wing primordium. Using CRISPR/Cas9 mediated deletions and lacZ reporter assays, the authors identified two conserved CRMs which function redundantly in a spatio-temporal manner and showed that these CRMs are also essential in the context of regeneration of the Drosophila wing.

While I think that the authors present interesting and important insights into wg1 enhancer-based regulation of the wg gene during development, regeneration and malignant growth, the paper (which is in most cases technically sound) currently lacks some clarity and fails at specific points to support their conclusions with the experiments performed. Major revisions are required before this comprehensive work can be published in Nature Communications. I understand that a lot of work has been done and thus only suggest additional experiments where I think that the the conclusions do not match the data presented.

We appreciate reviewer's comments on our work. We have been able to address all reviewer's concerns with additional experiments and changes in the text and in the figures (see below), which increase clarity and reinforce our conclusions.

Main comments

1) In lines 151-153 the authors state: "On the basis of sequence conservation with other Drosophila species, we subdivided a 5 kb-long region containing the wg1 deletion into four fragments [Alpha, Beta, Gamma, and Delta; Figure 3a] and generated reporter constructs carrying them." Throughout the manuscript the authors highlight and emphasize this conservation, however, they never show a sequence alignment or even report evidence that the characterized binding sites are part of these conserved elements. Moreover, the conservation has been the basis to subdivide the elements into the four fragments, they analysed and used in their experiments, two of which are stated to be the conserved regulatory modules (CRMs) and in fact the central message of the paper. The details how this comparison amongst Drosophila species was performed should be described in the Materials and Methods section.

In addition, the necessary sequence alignments (across species) as well as the conserved arrangement of the Ci, Ets, and AP-1 binding sites should be shown. Since the authors also perform experiments to show the relevance of the predicted binding sites and in part correlate their prediction score with their biological significance (high scoring being more relevant), also this aspect of conservation amongst different Drosophila species should be documented in the paper. In this respect, a Figure including the detailed information of all the fragments highlighting the position of the CRMS as well as the relative positioning of the individual TF binding sites (AP-1, Ci, ETS) within the CRM should be shown and labeled with an appropriate scalebar. Currently all this information is scattered amongst different Figures and Tables. Such a Figure would help the reader to get a better view on, and understanding of, the data. We appreciate reviewer's suggestion and agree that this information was not added to the first version of this ms. We apologize for that. We have now included a new supplementary figure (Figure S2) where we include the following data

- (1) Conservation of CRMs Alpha, Beta, Gamma and Delta by multiple alignments of 27 insect species (23 *Drosophila* species and *Anopheles*, *Apis*, *Tribolium* and *Mosca*) and measurements of evolutionary conservation using the USCS *Drosophila melanogaster* genome browser (<https://genome-euro.ucsc.edu/index.html>). A scale bar is also included.
- (2) Location of bioinformatically predicted Ci, ETS and AP1 binding sites in the *wg*¹ CRMs, their scores (according to the position weighted matrices and conservation) and their position weighted matrices.
- (3) Sequence conservation of Ci, ETS and AP1 binding sites by multiple alignments of 27 insect species and measurements of evolutionary conservation using the USCS *Drosophila melanogaster* genome browser (<https://genome-euro.ucsc.edu/index.html>).

We have included references to this figure throughout the ms as well as a new section in the Methods ("*Analysis of sequence conservation*", pg 19) to describe the methodology and genome browser that we used to analyze sequence conservation of the CRMs (and transcription factor binding sites) spanning the *wg*¹ enhancer.

Also, a "model" Figure summarizing all the data, findings and conclusions, i.e. signals/cues, the involvement of the enhancer element wg1 (particularly the role of the two independent CRMs) and the respective outputs Wg/Wnt6 with respect to development, regeneration and cancer would be very helpful to draw a take home message and to put the experiments into a bigger context. Especially the discussion would profit from such a figure. Without this information, the subdivision of the wg1 enhancer into two CRMs and two less relevant units (alpha delta) as well as the emphasis on conservation of CRM function, is difficult to grasp and remains vague. As suggested, we have included a new figure (Figure 8) where we summarize the data, findings and conclusions. We refer to this figure in the discussion section. We thank reviewer for this suggestion, as it will help the reader to better understand the take home message of our work.

2) In the Introduction (lines 26/27) the authors mention, that: "We identify two evolutionary conserved cis-regulatory modules within this enhancer that are utilized in a redundant manner to mediate these two activities through the use of distinct molecular mechanisms". While I agree that the mechanisms/signaling events that lead to enhancer activation are shown to be different, some of the experiments do not entirely allow to draw this conclusion. In the context of regeneration and malignant overgrowth, the authors delete one or both of the CRMs to study their function in this context. Their claim is that the responses monitored are mediated by the AP-1 binding sites within the two CRMs. However, the monitored responses are reductions and not complete eliminations of Eiger-induced wg expression (lines 264-266; "The levels of Wg expression induced by Eiger were significantly reduced in Δ BetaGamma homozygous flies and partially reduced in flies where either the Gamma or Beta CRMs were deleted (Figure 6c, d)", and thus it may be that slightly lower scoring AP-1 binding sites (as depicted in Table 2) remain in the alpha fragment (i.e., 3rd CRM?).

We completely agree with the reviewer that other AP1 binding sites besides the ones identified in these two CRMs might be responding to JNK as stated in pg11 of the ms in the following sentence (now marked in red in the revised ms): "*Interestingly, Eiger was able to cause ectopic expression of Wg even upon deletion of these two CRMs, thus indicating the presence of other genomic regions acting on Wg and responding to JNK.*"

In order to address reviewer's comment on the contribution but not absolute requirement of the AP1 binding sites present in these two CRMs, we have included the underlined change in the sentence of lines 26/27 of the abstract: "*We present evidence that a single wing-specific enhancer in the wingless gene is sequentially used in two*

consecutive developmental stages to drive wing specification first and then contribute to mediating the remarkable regenerative capacity of the developing wing upon injury."

In addition, the authors also delete all binding sites for ETS and Ci and possible other, so far unknown regulatory elements by deleting the entire CRMs. In fact, the authors mention (line 56ff) that "...in older larvae and once wing specification is underway, this enhancer remains silent but accessible to transcription factor binding in growing wing primordia". Therefore the authors cannot completely rule out that the observed effects are exclusively due to the elimination of AP-1 sites. To really prove that the AP-1 binding sites themselves are necessary and sufficient for the effects they observe, the authors should perform the following experiments: They should repeat a subset of the regeneration and malignant overgrowth experiments making the following changes. Rather than assessing Wg expression in the Δ BetaGamma deletion setting, the authors should use wg1-lacZ reporter where exclusively (some of) the AP-1 sites are mutated, in either or both of the CRMs leaving the rest of the CRMs intact. The responses can be compared to the wild-type wg1-lacZ transgenes already available. As suggested by this reviewer, we have analyzed the impact of mutating four AP-1 sites of the wg1 enhancer (see Methods section for details) on the expression levels of the lacZ reporter in regenerating and overgrown wing discs when compared to the available wild-type wg1-lacZ reporter. As shown in Figures 6b and 7d, the expression levels of the lacZ reporter carrying the mutated version are shown to be reduced in the two experimental settings when compared to a control reporter carrying the wild type version of the wg1-enhancer. Data have been discussed in the text as follows

Pg 11: "The ability of the wg1 enhancer to drive lacZ expression upon Eiger expression was drastically compromised when mutating four out of the five existing AP1 binding sites (Figure 6b; mutated AP1 binding sites are labelled with a star in Figure 6a)."

Pg 13: "Mutating four out of the five existing AP1 binding sites (labelled with a star in Figure 6a) in the wg1 enhancer compromised its ability to drive lacZ expression in CIN tissues (Figure 7c).

Also, the authors might consider making AP1 deletions (with and without Ci deletions) in the context of wg1-lacZ reporter transgene – that way the authors could better distinguish between the role of these CRMs (and their redundancies) in development (Hh signaling, early expression and maintenance of Wg) vs regeneration / stress (CIN cancer model).

In order to distinguish between the roles of these CRMs in development vs regeneration/stress, we have analyzed the developmental expression of the wg1 enhancer carrying mutations in four out of the five existing AP1 binding sites. As shown in Figure S4e, its restricted expression to a ventral anterior wedge is not affected. This has been discussed in the ms as follows in pg11: "We observed that the developmental expression of the wg¹ enhancer was largely unaffected by these mutations (Figure S4e)."

Minor comments

In lines 125-127, the authors state that "Surprisingly, we observed that expression of wg1-lacZ was not abolished upon Ptc overexpression (Figure 2g, arrows), suggesting that Hh signalling is not an absolute requirement for the expression of Wg in early wing discs." While this is a very likely scenario, an alternative explanation may be possible: It could be, that the ptc overexpression via sd-Gal4 cannot completely abolish wg1-lacZ expression because the scalloped expression domain may not entirely cover all the cells of the wg-lacZ expression domain in the second instar larvae. Unfortunately I did not find this data in the manuscript. Even after a literature search it remains unclear how sd is expressed in a 1st or 2nd instar wing disc. Importantly, this should be shown in the paper as the function of the wg1 enhancer is studied in this context and not the

3rd instar larvae. Given the dynamic changes of *wg* expression in 1st, 2nd and 3rd instar larvae *sd-Gal4* does not necessarily completely overlap with the *wg*-expression domain.

We have previously used this Gal4 driver to unravel a role of Notch (Rafel et al 2008) and JAK/STAT signaling (Recasens et al 2017) pathways in wing fate specification. As shown in Rafel et al 2006 (Figure 1C), this Gal4 driver is ubiquitously expressed in the whole wing primordium of second instar larvae. We have now included in Figure 2h of the present manuscript two examples of early second and early third instar wing discs where *sd-gal4* is shown to be ubiquitously expressed in the whole wing primordium. We have referenced these data in the ms as follows in pg6: “We next genetically manipulated *Hh* expression and signaling with the use of the *sd-gal4* driver, which is ubiquitously expressed in the early wing primordia (Figure 2h), and addressed the impact on the expression of the *wg*¹-*lacZ* reporter.”

In order to reinforce the role of *Vein* in restricting the expression of the *wg*¹-enhancer to the most ventral part of the wing disc, we have included as Figure 2i an example of a second instar wing disc where *vn-argos* is expressed under the regulation of the *sd-gal4* driver.

Confusing is also the labelling of Figure 2g: In the Figure legend it is mentioned that the larvae expressed the indicated transgenes under the control of the *sd-gal4* driver. However, the panel is entitled *sd>EGFP* the green staining should however show *Ci* expression. Please clarify the labelling of the Figure.

We have corrected the labeling to avoid confusion.

Similarly to the *sd-Gal4* expression domain, the authors should shortly explain in the text why they are using *rd-Gal4* constructs and how they are expressed, as the paper should be easily accessible also to a more broader audience that does not have a *Drosophila* background. The reader should not be forced to read all the previously published papers to figure out that this line has been used and described elsewhere (i.e., Ref 13).

We appreciate this comment as it increases clarity to the non-expert. We have included the following sentences (underlined and labeled in red in the ms) in pg 11 to explain where *rotund-gal4* and *spalt-lexA* drivers are expressed.

“Overexpression of the pro-apoptotic gene reaper was used in ³¹ to directly induce apoptosis. In both cases, transgene expression was driven by the *rotund-gal4* (*rn-gal4*) driver, which is expressed in those cells that will give rise to the adult wing (Figure 5b, region coloured in yellow). The GAL4/UAS system was combined with the temperature-sensitive version of the Gal4 repressor Gal80 (*Gal80^{ts}*) to drive transgene expression during a discrete period of time in early third instar wing discs and to analyse the capacity of the remaining tissue to give rise to a normal adult wing as a consequence of compensatory proliferation. We used two different transactivation systems (*Gal4/UAS* and *LexA/LexO*) and shortened the induction period to 11 h (in the case of *Reaper*) or 16 h (in the case of *Eiger*) in early third instar larvae (Figure 5b). We used *m-gal4* to drive *Eiger* expression and *spalt-lexA*, which is expressed in a central region of the presumptive wing (Figure 5b, region coloured in brown), to drive *reaper* expression.”

Finally, the authors may consider changing the current title “ Sequential activities of a WNT enhancer in specification and regeneration of the *Drosophila* wing”. Sequential implies that the events follow one another which must not necessarily be the case. On the other hand, both events (specification and regeneration) may accidentally happen at the same time - which may provoke the mentioned discrepancies in the findings (mentioned in line 237-239). For this reason, the authors refine their experimental settings (lines 252-254). I believe that the current title is not well chosen.

We have changed the title to the following one: "*A single WNT enhancer drives specification and regeneration of the Drosophila wing*" where the words "sequential activities" are being removed, as suggested by this reviewer.

REVIEWERS' COMMENTS

Reviewer #1 (Remarks to the Author):

The authors have addressed all my previous concerns. The text/figure changes and the newly performed experiments are of high quality and answer satisfactorily all the questions risen during the review process. Of special note is the new Figure 8 containing a very clear model summarizing the findings of the manuscript. I appreciate the thorough response in the rebuttal letter concerning the Wg regulation by Wnt6 and I encourage the authors to pursue this direction of research.

To conclude, I have no additional comments on the manuscript and I recommend its publication.

Salvador Herrera, PhD.

Reviewer #2 (Remarks to the Author):

With the revised manuscript, the authors have addressed all reviewer's comments appropriately. I therefore support publication in Nature Communications.

I would still have a minor comment though:

May I ask the authors to avoid the wording: "enhancer is expressed..." throughout the text as an enhancer (or a CRM) is a regulatory element that guides, influences, drives or varies expression of genes, but usually is not expressed itself. If part of an intron it will be transcribed but this "expression" is not part of its function per se. Examples are in the main text line 235 "Although Alpha and Gamma were expressed in the leg discs..." or Title of Figure S3 "The wg1-enhancer is expressed in other tissues".

line 435 typo : "were used toi generate.." should read "were used to generate."

REVIEWERS' COMMENTS

Reviewer #1 (Remarks to the Author):

The authors have addressed all my previous concerns. The text/figure changes and the newly performed experiments are of high quality and answer satisfactorily all the questions risen during the review process. Of special note is the new Figure 8 containing a very clear model summarizing the findings of the manuscript. I appreciate the thorough response in the rebuttal letter concerning the Wg regulation by Wnt6 and I encourage the authors to pursue this direction of research.

To conclude, I have no additional comments on the manuscript and I recommend its publication.

Salvador Herrera, PhD.

We appreciate reviewer's comments on our paper and appreciate his great input throughout the reviewing process.

Reviewer #2 (Remarks to the Author):

With the revised manuscript, the authors have addressed all reviewer's comments appropriately. I therefore support publication in Nature Communications.

I would still have a minor comment though:

May I ask the authors to avoid the wording: "enhancer is expressed...." throughout the text as an enhancer (or a CRM) is a regulatory element that guides, influences, drives or varies expression of genes, but usually is not expressed itself. If part of an intron it will be transcribed but this "expression" is not part of its function per se. Examples are in the main text line 235 "Although Alpha and Gamma were expressed in the leg discs..." or Title of Figure S3 "The wg1-enhancer is expressed in other tissues".

line 435 typo : "were used toi generate.." should read "were used to generate."

We appreciate reviewer's comments on our paper and appreciate his/her great input throughout the reviewing process. We have addressed the minor comment as well as the typo of line 435. All changes are highlighted in red in the ms and SI documents.